# Proteomics of Multiple Sclerosis: Inherent Issues in Defining the Pathoetiology and Identifying (Early) Biomarkers

**DOI:** 10.3390/ijms22147377

**Published:** 2021-07-09

**Authors:** Monokesh K. Sen, Mohammed S. M. Almuslehi, Peter J. Shortland, David A. Mahns, Jens R. Coorssen

**Affiliations:** 1School of Medicine, Western Sydney University, Locked Bag 1797, Penrith, NSW 2751, Australia; monokesh.sen@westernsydney.edu.au (M.K.S.); m.almuslehi@westernsydney.edu.au (M.S.M.A.); d.mahns@westernsydney.edu.au (D.A.M.); 2Department of Physiology & Pharmacology, College of Veterinary Medicine, University of Diyala, Baqubah 32001, Diyala, Iraq; 3School of Science, Western Sydney University, Locked Bag 1797, Penrith, NSW 2751, Australia; p.shortland@westernsydney.edu.au; 4Departments of Health Sciences and Biological Sciences, Faculties of Applied Health Sciences and Mathematics & Science, Brock University, St. Catharines, ON L2S 3A1, Canada

**Keywords:** proteoforms, bioinformatics, cuprizone, experimental autoimmune encephalomyelitis, top-down proteomics, protein species, post-translational modifications, bottom-up proteomics, neurodegenerative disease

## Abstract

Multiple Sclerosis (MS) is a demyelinating disease of the human central nervous system having an unconfirmed pathoetiology. Although animal models are used to mimic the pathology and clinical symptoms, no single model successfully replicates the full complexity of MS from its initial clinical identification through disease progression. Most importantly, a lack of preclinical biomarkers is hampering the earliest possible diagnosis and treatment. Notably, the development of rationally targeted therapeutics enabling pre-emptive treatment to halt the disease is also delayed without such biomarkers. Using literature mining and bioinformatic analyses, this review assessed the available proteomic studies of MS patients and animal models to discern (1) whether the models effectively mimic MS; and (2) whether reasonable biomarker candidates have been identified. The implication and necessity of assessing proteoforms and the critical importance of this to identifying rational biomarkers are discussed. Moreover, the challenges of using different proteomic analytical approaches and biological samples are also addressed.

## 1. Background

### 1.1. Multiple Sclerosis

The blood–brain barrier (BBB) separates the central nervous system (CNS) from other bodily systems, including the peripheral immune system [1,2]. In clinically confirmed Multiple Sclerosis (MS) patients, the BBB is compromised, allowing free access of adaptive immune cells (e.g., T-cells, B-cells) into the CNS [3,4,5]. In the CNS, these immune cells contribute to a complex chronic neuro-inflammatory state, oligodendrocyte degeneration or death (i.e., oligodendrocytopathy or oligodendrocytosis), demyelination and neuronal loss [6,7,8]. However, other biochemical changes in the CNS may well precede the involvement of the peripheral immune system [9,10,11,12,13]. Based on the clinical diagnostic criteria, MS is categorized into four broad phenotypes: primary progressive MS (PPMS), secondary progressive MS (SPMS), relapsing-remitting MS (RRMS), and progressive relapsing MS (PRMS; [7,14,15]). Although all MS phenotypes show demyelination and inflammatory episodes, the pattern of these episodes is largely heterogeneous in the early phases of the disease and becomes more homogenous over time [16]. 

### 1.2. MS Diagnosis and Biomarkers

The current standard procedure for MS diagnosis is based on the detection of two demyelinating events separated in time and space using magnetic resonance imaging (MRI; [17,18]). Since MS diagnosis includes the assessment of clinical symptoms (e.g., sensory-motor deficits) and MRI detection of new and/or recurrent lesions, understanding the initial or initiating event(s) (i.e., pathoetiology) using MRI is improbable. Likewise, changes in metabolites have been found prior to the detection of myelin damage/demyelination using MRI, suggesting that MRI is not sensitive to earlier events [19]. For example, a 30–40% increase in specific metabolites (e.g., choline) was found (using proton magnetic resonance spectroscopic imaging) in the brain before MRI detection of lesions in normal-appearing white matter [19]. Additionally, metabolic changes (e.g., reduction of N-acetylaspartate) were found in the brain areas of MS patients and correlates with disability in which conventional MRI image analysis was unable to show a correlation [20,21]. Likewise, whole-brain proton magnetic resonance spectroscopy revealed a significant reduction of N-acetylaspartate in patients with clinically isolated syndrome (having neurological disability similar to MS before a second episode of demyelinating lesions) [21,22]. In contrast, no demyelinating lesions were found using standard MRI protocols [21]. N-acetylaspartate, a metabolite of aspartic acid, is highly expressed in neuronal and glial cells and its reduction indicates axonal injury and is correlated with disease severity in RRMS patients [20,23]. Consistent with the early metabolic changes preceding clinical diagnosis of MS [21], a recent report using Nile Red fluorescence spectroscopy revealed subtle biochemical alterations in the lipid constituents (i.e., metabolic changes) of histologically intact (i.e., pre-lesion) medial corpus callosum after only two days of feeding cuprizone (CPZ) to mice [13]. A similar change was observed in normal-appearing white and grey matter from MS patients [13]. Notably, using traditional histological staining methods (e.g., Luxol-fast blue), it takes 2–5 weeks of CPZ-feeding to detect measurable changes [24]. Thus, the detection of myelin changes is method-specific and traditional histological methods do not reveal early or subtle biochemical abnormalities associated with disease onset and thus etiology. Moreover, the progressive neuroinflammation (e.g., glial activation) by MRI measurement is limited but requires alternative techniques such as positron emission tomography (PET) of radio-labelled translocator protein (TSPO; [25,26,27]). However, this method, like MRI, is technologically demanding and expensive [28,29]. Furthermore, the quality of TSPO radioligands and post-processing methodology need further improvement (e.g., overcoming interference by radio labelled metabolites during TSPO signal quantification) if it is to become a reliable, routine, diagnostic method [26]. 

In addition, one of the earliest clinical indications of BBB disruption in MS is the presence of gadolinium-enhanced lesions (detected by MRI) and this disruption is associated with the presence of neuroinflammation at the lesion site [30]. Importantly, in RRMS patients, the greater the BBB disruption (measured as the constant K^trans^ using three-dimensional dynamic-contrast enhanced MRI), the greater the inflammation (measured by quantifying urinary neopterin, a product of activated macrophages) [31]. Although the BBB breach is considered transient, the severity of the disease (e.g., mild or relapsing-remitting) appears to be dependent upon the magnitude of the BBB permeability. MS patients with more lesions (indicative of greater BBB disruption) have a higher frequency of symptoms [32]. However, BBB disruption occurs in other demyelinating diseases (e.g., neuromyelitis optica) as well as neurodegenerative diseases [33,34]. However, routine monitoring of BBB disruption in MS (and other demyelinating diseases) using MRI would likely be prohibitively expensive. 

Additional, but not obligatory, confirmation of MS includes the detection of non-specific oligoclonal immunoglobulin G (~90% of MS patients) in cerebrospinal fluid (CSF; [6,35,36]) indicative of immune system involvement [37,38,39]. However, oligoclonal bands are not exclusive to MS patients [40]. Elevated levels of immunoglobulin G are detected in patients with other diseases such as systemic lupus erythematosus, Alzheimer’s disease [41,42,43], neurosyphilis [44], subacute sclerosing pan-encephalitis [45], and falciparum malaria [46]. It is also not unusual for patients to be diagnosed with MS in the absence of oligoclonal bands [47,48]. Notably, a significant number of patients are also routinely misdiagnosed as either positive or negative for MS [49,50]. Validated protein biomarker(s) would thus provide a unique diagnostic ‘molecular fingerprint’ specific to MS; such biomarkers would have their greatest utility if providing the earliest possible detection, even before the appearance of early clinical symptoms (e.g., sensory-motor deficits, vision problems), which are distinctly associated with disease progression [51]. How such early biomarkers could be broadly implemented beyond known familial cases of MS remains to be determined and would likely require a broadly concerted and targeted effort to stamp-out the disease. Such biomarkers would thus also be used to measure the efficacy of therapeutics [52,53,54]. 

‘Omics’ approaches provide promise to define critical molecular alterations, although the significance of their impact hinges entirely on what is sampled and when, the analytical method, as well as the nature of the model systems available [55,56]. If proteomic biomarkers are identified from reasonably non-invasive, easily collected samples (e.g., blood or urine), it could lead to earlier diagnosis, prognosis, and therapeutic intervention, with better sensitivity and a markedly lower cost than current imaging modalities. This review thus investigated whether currently identified protein species can be considered as biomarkers for MS, using comprehensive literature mining and bioinformatics to anchor the analysis. During the last two decades, proteomic approaches were used to assess various samples from animal models of MS (e.g., experimental autoimmune encephalomyelitis, EAE and cuprizone, CPZ), as well as from MS patients [57,58,59,60,61,62,63,64,65,66,67,68,69,70,71,72,73,74,75,76,77,78,79,80,81,82,83,84,85,86,87,88,89,90,91,92,93,94,95,96,97,98,99,100,101,102,103,104,105]. However, there do not appear to be any critical systematic reviews of the protein species identified from these studies. Previous reviews [106,107,108,109] were far more selective in their focus (e.g., concerning sample type, method used), and none discussed critically the different key factors involved in proteomic analyses (e.g., sample collection, protease inhibitor use, and data analysis). Importantly, none critically discussed proteoforms, the active protein species in any given biological function [110,111,112,113]. Thus, another primary goal of this review was to critically evaluate published studies from 2004–2019 that used proteomic approaches to assess MS patient samples, as well as CPZ and EAE models, in order to identify key molecular pathways associated with the identified canonical proteins, and the potential for these to be effective biomarkers for MS. The review addresses the central importance of protein species (i.e., proteoforms) in identifying unique and disease-specific biomarkers. We suggest guidelines that should be considered in future proteomic studies. We believe this review to be of broad academic interest beyond MS research and are hopeful that it will promote a wider understanding of the importance of analyzing and identifying proteoforms rather than canonical protein sequences in identifying critical biomarkers. Additionally, it will promote a deeper and more consistent understanding of the mechanisms underlying neurodegenerative disease states, including MS. Consensus in analytical design and methodological approaches will be the key to critical future advances. 

### 1.3. Proteomic Analyses

In the broadest terms, proteomic analyses assess the variety and abundance of ‘proteins’ in body fluids, cells, tissues, and organs [114,115]. However, the identification of canonical proteins (i.e., simply based on amino acid sequence) is now widely recognized as an insufficient approach considering the complexity of proteomes [110,111]. While the number of genes is unchanged in a cell or organism, the proteome is highly dynamic, responding to all physiological and environmental changes [116,117,118]. Proteoforms originate from complex processes such as alternative RNA splicing and post-translational modifications (PTM, e.g., phosphorylation, glycosylation, acetylation, proteolytic cleavage). The estimated number of human proteoforms is ≥1 million and a given proteome is thus far greater in size and more complex than the genome (~20,000 genes) or transcriptome [111,119,120,121]. The diversity and functionality of proteoforms are vast as they account for all known ‘protein’ functions; the range of proteoforms can thus also vary substantially between cell types and organisms [122]. This complexity can simply no longer be ignored in favour of ‘fast’ assays that provide routine identification of only canonical proteins (from here on referred to generically as ‘proteins’ except where proteoforms were identified via molecular weight (MW), isoelectric point (pI), or PTM information using top-down analyses). While proteoform detection, identification, and quantification is complicated, their linkage to specific molecular functions is critical in terms of understanding molecular mechanisms (e.g., pathophysiology) and for the identification of the best possible biomarkers and drug targets [123]. However, there are limited approaches available to address and understand the complexity of proteomes, recognizing that proteomes are composed of proteoforms rather than simply canonical amino acid sequences [110,111,113,124]. 

Many studies have revealed the association of proteoforms in the pathogenesis of MS. Kin et al. found an elevation of mono and dimethylated arginine but a reduction of phosphorylation in MS white matter samples [125]. Consistent with this, deimindaton, or citrullination, results in a conformational change when the amino acid arginine in myelin basic protein is converted to citrulline, a non-standard amino acid; this has been reported in MS lesions [126]. In contrast, these modifications were not found in non-MS human subjects [125,126]. Similar to the detection of this PTM in samples from human MS patients [125,126], citrullination has also been associated with autoimmune encephalomyelitis in the CPZ [10] and EAE animal models [127]. Notably, citrullination has also been linked to rheumatoid arthritis [128], and there is also evidence of an increase incidence of rheumatoid arthritis in MS patients [129]. Specific PTM have also been associated with other neurological diseases such as Alzheimer’s disease, and with traumatic brain injury [130,131]. In the latter case, following traumatic brain injury, tau acetylation resulted in neurodegeneration and neurobehavioral impairment in an animal model; inhibition of acetylation reversed the pathological outcome [131]. While far from exhaustive, these examples highlight that proteoforms are the functional, biologically active molecules, and emphasize the critical need to identify specific proteoforms in order to understand the native molecular mechanisms and thus also the pathophysiology of a disease. 

Currently, two general approaches, called top-down and bottom-up, are used for quantitative proteome analysis (Figure 1). Top-down analyses are based on either mass spectrometry-intensive or integrative approaches [110,124]. The former initially involves lower resolution gel-based separation of proteins (largely employing the GELFrEE technique [132,133]) and liquid chromatography-tandem mass spectrometry (LC-TMS) to identify intact species. This mass spectrometry-intensive approach currently provides detailed identification of intact species, but this is largely only in the ~10–50 kDa range. In contrast, the integrative approach is based first on highly refined two-dimensional gel electrophoresis (2DE), in which proteoforms are initially separated on the basis of their charge (pI) and then resolved by sodium dodecyl sulfate (SDS)-polyacrylamide gel electrophoresis (PAGE) according to their size (i.e., relative MW). This is then coupled with LC-TMS (i.e., 2DE/LC-TMS) to potentially identify thousands of proteoforms from any biological sample, enabling the deepest routine proteome coverage currently available [104,112,113,134,135,136]. In contrast to the top-down approaches that directly resolve intact proteoforms, in bottom-up, or so-called ‘shotgun’ studies, protein extracts are first subjected to gross protease digestion to obtain a highly complex peptide mixture. This is then analysed using LC-TMS and established online protein databases to infer the canonical proteins (i.e., amino acid sequences) that are potentially present in the sample [137]. Without extensive, exhaustive, and labour-intensive separate selective analyses, all information concerning proteoforms is lost. 

The shotgun approach is nonetheless claimed to provide a relatively simpler and less time-consuming analytical approach compared to top-down analyses [137,138]. This may be true if technical replicates and an exhaustive series of separate analyses for all known PTM are not carried out. Therefore, while the shotgun approach has gained substantial popularity [87,105,137,138,139,140], top-down analyses are more accurate and thorough in the assessment of the relevant, biologically active species, the proteoforms [110,113,124,141]. Literature mining and utilization of bioinformatic tools (e.g., DAVID, https://david.ncifcrf.gov/, PANTHER, http://www.pantherdb.org/ and STRING, https://string-db.org/ [104,140]) are then used to begin the process of assessing the potential biological role(s) and interactions of the identified proteoforms. It should be noted, however, that these databases tend to generically address protein functions as found in the literature rather than the functions/interactions of specific proteoforms [87,104,142]. 

## 2. Proteomic Investigations into MS

Assessment of proteome profiles in MS began in 2004 [63,64], although targeted detection of oligoclonal bands in CSF was initiated in the 1970s using isoelectric focusing [143,144,145]. Subsequent studies used one-dimensional gel electrophoresis to detect oligoclonal bands in CSF and blood [146,147]. Proteome mapping using high-resolution 2DE of MS samples was introduced approximately two decades ago to resolve CSF proteomes from RRMS patients and showed changes in 65 [63] or 61 proteins [64], respectively (based on ≥2 matching peptides). Both studies revealed consistent changes in the abundance of the several same proteins, including structural (e.g., actin and gelsolin), blood-related (e.g., transthyretin, haptoglobin, and pigment epithelium-derived factor), metabolic (e.g., prostaglandin D2 synthase), and immune complement species. The consistency of these early results between studies using 2DE, despite the use of different stains (e.g., silver [63] or Coomassie Brilliant Blue (CBB; [64]), attests to the reproducibility of this method. For the EAE model, the first published proteomic assessment was in 2005 using cerebral micro vessel samples resolved by 2DE and identified changes in the abundance of five proteins [86]. These were not found in MS samples [63,64]. Regarding the CPZ model, the shotgun approach was first used to assess the cerebrum proteome, identifying changes in the abundance of 70 canonical proteins [105]; however, only a few proteins such as actin and apolipoprotein were common to MS proteome samples [63,64]. Following these initial studies, there have been many proteomic analyses using samples from MS patients and animal models of MS (EAE and CPZ), reflecting the importance of proteomics in MS research, and more broadly in the analysis of disease processes [57,58,59,60,61,62,65,66,67,68,69,70,71,72,73,74,75,76,77,78,79,80,81,82,83,84,85,87,88,89,90,91,92,93,94,95,96,97,98,99,100,101,102,103,104,105]. 

## 3. Search Strategy and Selection Criteria

A PubMed search (https://www.ncbi.nlm.nih.gov/pubmed/) was performed by combining the terms MS and proteomics (top-down and bottom-up), an animal model of MS, cuprizone (CPZ) animal model and proteomics, experimental autoimmune encephalomyelitis (EAE) animal model, and proteomics. In total, 49 proteomic studies published between January 2004 and October 2019 were initially evaluated, and these were then updated with more recent studies prior to publication. ‘Uncategorized/uncharacterized putative’ canonical proteins in the list were not considered for further analysis. Likewise, if the same protein was observed in multiple samples or protein spots in the same study, a single representative protein was used for analysis. Protein alterations found in the remyelination phase or when a drug was tested for remyelination in the CPZ model were also excluded from the analysis [102,105,148]. Moreover, if only a single peptide was used as the basis for canonical protein identification, that identification was also discarded [63,75,77]. Of six different animal models (e.g., EAE, CPZ, ethidium bromide, lysolecithin, diphtheria toxin and virus) [14,149,150,151,152], proteomic studies were only found for EAE and CPZ, and these were included in the analysis. Protein accession IDs were uploaded to the UniProt website (www.uniprot.org; accessed in January 2020) to obtain gene ID and canonical protein names.

## 4. Protein Biomarkers and Biological Samples

Biomarkers are indicators and predictors of particular (patho)physiological processes. Therefore, a validated set of proteoform biomarkers would aid in early definitive diagnosis and prognosis, identification of molecular pathways underlying disease initiation and progression, as well as complement existing clinical tools in assessing the effectiveness of therapeutics. From a clinical standpoint, one of the critical steps in identifying biomarker candidates is the selection of biological samples, obtained by minimally invasive methods. Blood is often the first choice because it is easily accessible and contains both cellular (e.g., white blood cells) and fluid components (e.g., serum or plasma; [153]). It has been noted that there is a high percentage (~80%) of overlapping canonical proteins between CSF and blood, suggesting that blood can be a viable alternative biological sample for MS patients [52,153,154]. Additionally, urine and CSF are also routinely sampled for diagnostic purposes [52,109,155]. Of the 29 proteomic studies that assessed MS patient samples, 19 studies were found using CSF [59,60,61,63,64,65,66,67,68,69,70,71,72,73,75,77,78,82,83], whereas five studies were found using blood [57,62,74,79,85] samples. One study was found using both serum and CSF samples [84], and two studies used post-mortem CNS cerebral tissue samples (e.g., cerebrum; [58,76]), indicating that CNS tissue is infrequently used for proteomic analyses in MS. Since urine is a blood filtrate, it may also contain potentially critical biomarkers [156,157,158]. Likewise, tears [80,159] and saliva [160,161] also contain proteins that might be potential biomarkers for MS. Although urine, tears, and saliva are easily accessible, urine and tears have been rarely used [80,81] or, in the case of saliva, not used at all in MS proteomic studies. We propose that these samples should be more extensively studied in MS research to identify potential proteomic biomarkers (Table 1). 

Notably, biomarkers may, or may not, be directly involved in a given (organ-specific) process; that is, while they may have strong diagnostic/prognostic capacity, they may not reflect critical underlying cellular/molecular mechanisms. Thus, regardless of where they arise, or what fluid they are found in, the point is whether current approaches and the proteins identified can serve as effective biomarkers. Other critical considerations also include potential further processing of putative biomarkers. For example, while elevated serum neurofilament light chain is considered a promising marker of neurodegeneration or CNS trauma [162,163], this elevation can be confounded by altered renal function in MS patients. Elevated serum neurofilament correlates with elevated creatinine levels, suggesting impairment of renal function in older adults with neurodegeneration [164]. This is supported by the findings of Calabresi et al. [165], showing a reduction of glomerular filtration rate in patients with progressive MS. This suggests that regular monitoring of renal function be recommended. 

In CPZ studies, proteome analyses have mainly focused on cerebral tissue (Table 1) as demyelination and consequent glial activation occurs in the brain [101,102,103,104,105]. Other samples from the CPZ model (e.g., peripheral blood mononuclear cells and splenic tissue) have also been assessed [103]. Recently, we [166] and others [167,168] showed that there is marked demyelination and gliosis in the cerebellum and brain stem of CPZ-fed mice. In contrast, in the spinal cord, there was glial activation without demyelination [166]. However, potential protein alterations have not been tested in any of these regions to investigate the temporal effect of CPZ on proteome profiles. EAE proteome assessments have mainly focused on the spinal cord (Table 1) because, in contrast to MS, it is the main CNS region affected in this model [88,89,93,94,96,97]. Cerebral [90] and brain stem tissue [100] or serum [98] have been less commonly used as samples for analysis in EAE. However, whole CNS tissue was used only in one study [92]. Clearly lacking is a detailed proteomic analysis of CNS tissue in both CPZ and EAE animals. Likewise, apparently no proteomic studies have examined the cerebellum, brain stem, or spinal cord from MS patients. Notably, as shown in Table 1, blood is the only sample that was analysed in the three different biological systems (i.e., MS, EAE, and CPZ) to compare protein changes across live individuals. The cerebrum was analysed in post-mortem samples in each of the systems, with the caveat, of course, being that the animal models were sacrificed at defined time points (Table 1). 

Since in most MS patients the BBB is (or was) disrupted, the proteinaceous CNS material that is released into the circulation can be assessed by proteomic analysis and may represent potential biomarkers. In this regard, the search for potential biomarkers via proteome analysis of CSF samples might be more informative than blood, as drainage of the proteins into the CSF more directly reflects the CNS pathological status. Despite its far lower protein concentration relative to blood, most of the CSF protein content is CNS-specific or of very low abundance elsewhere in the body [106]. Therefore, while CSF may seem the ‘best’ choice, there are other factors (e.g., collection, contamination) to be taken into account before considering CSF-based biomarker discovery. CSF collection requires highly experienced personnel and special care during lumbar puncture and is associated with potential side effects including headache and pain. Furthermore, it would likely be difficult to collect enough age/gender-matched control samples [169,170], and it would be difficult, if not impossible, to institute serial sampling of the sort required for the best possible identification of early diagnostic and prognostic markers. Furthermore, it is not always feasible to collect CSF, especially from undefined presymptomatic individuals and from children [79]; in contrast, blood or urine can be collected routinely to use for early protein biomarker discovery. However, CSF can also be contaminated by blood-borne proteins [171,172]. A study that analysed CSF and CSF spiked with whole blood samples found four highly abundant protein species including hemoglobin, catalase, peroxiredoxin, and carbonic anhydrase I [171] indicating the contamination of CSF with blood. Whether or not the changes in blood-related proteins in the CSF or CNS tissue in MS studies result from sampling/handling procedures remains undefined. In this regard, it is also noteworthy to mention that most of the proteomic studies reviewed here have only listed the identified canonical proteins, without further validation or identification of relevant proteoforms, and are devoid of independent replication. 

In human MS research (or other diseases such as Alzheimer’s or Parkinson’s), the use of CNS tissue samples is limited, and these can only be collected post-mortem. Although there are a number of notable complicating factors in sampling post-mortem, not least of which is the time between death and autopsy, proteomic assessment of such samples provides only information concerning the disease status at the time of death. Since death would normally be after disease development and prolonged ill health (along with other potential comorbidities such as cardiovascular disease and diabetes; [173]), as well as likely prolonged use of a variety of medications, any post-mortem molecular analyses are unlikely to reveal anything definitive concerning the pathoetiology. Thus, for proteomic analysis of CNS disorders (including MS), blood, urine, or even saliva samples should be used routinely rather than CNS tissue or CSF for proteome-based biomarker discovery [81,154,174,175]. The current review found that 22% of the shared canonical proteins have contradictory trends (i.e., increase in some studies vs decrease in others), which were also shared between CSF and blood (Table 2 and Table 3). Only 18% of studies used blood samples, whereas 68% used CSF samples from MS patients (Table 1). The highest number, ~80%, of these common proteins were found in CSF samples (Table 2). Importantly, 67%, 38%, and 19% of these proteins showed comparable changes in blood, tears, and urine, respectively, as summarised in Table 2. These data again suggest that these alternate, easily accessible samples can be considered as viable alternatives for MS proteomic investigations, particularly for preclinical stages of the disease (see below). 

## 5. Other Factors Affecting Proteomic Analyses

### 5.1. Analytical Approaches

Following initial use of the top-down approach [63,64] in MS research, there was a gradual increase in the use of shotgun analyses [57,58,60,67,69,70,73,74,76,77,81,82,84,85]. However, the routine bottom-up/shotgun approach requires less sample than top-down but only infers canonical protein identifications, providing no information regarding proteoforms. The shotgun approach thus involves bulk digestion of a total protein extract followed by mass spectrometric analysis (Figure 1). In contrast, the top-down proteomic approach uses two steps to resolve intact proteoforms prior to selective proteolytic digestion and mass spectrometric analysis (Figure 1). Therefore, while the bottom-up approach is often claimed to be ‘faster’, this is often by ignoring technical replicates and does not take into account the many multiple separate analyses that must occur in this approach for every potential PTM if critical information concerning proteoforms is even sought [110,113,124]. Nonetheless, in what can hopefully be seen as a comment on the complementarity of available approaches, for MS samples, 14/29 articles employed top-down methodologies [59,61,62,63,64,65,66,68,71,72,75,78,79,80] while 15 used bottom-up approach [57,58,60,67,69,70,73,74,76,77,81,82,83,84,85]. Of the 5 CPZ studies reviewed, three used bottom-up [101,102,105] and two used a top-down approach [103,104]. In EAE studies, 8/15 were top-down [86,88,89,90,91,95,96,100] and the rest were bottom-up [87,92,93,94,97,98,99]. Thus, the data indicate that relatively equal numbers of top-down and bottom-up analyses have been employed across MS, CPZ, and EAE proteomic studies, although never in the same lab with the same samples. Rigorous sampling and analysis protocols of the same sample types, at different sites, is a key to biomarker discovery and to rule-out lab-to-lab variabilities [177]. However, a marked difference was observed when the number of identified species was tallied from these two approaches (top-down vs bottom-up). For example, a major difference was found in EAE, with ~10% of proteoforms reported from top-down studies but ~90% were canonical protein identifications from bottom-up assessments. Likewise, 43% and 46% of hits were from top-down, while 57% and 54% were identified using the bottom-up approach for CPZ and MS samples, respectively (Appendix A). The dominance of identifications from the bottom-up strategy was also observed in the common changes (i.e., changes in at least two biological systems; e.g., MS and EAE, irrespective of sample types, e.g., blood, CSF) summarised in Table 3. It was found that ~76% of canonical proteins showed comparable changes (irrespective of sample type) in bottom-up analyses; the rest were proteoforms found using the top-down approach (Table 3). The likely reason that more canonical proteins were identified by the bottom-up approach is that proteoform identifications in rigorous top-down studies demand that several criteria be satisfied. These criteria often include (i) 100% detectability in every technical replicate of every sample; (ii) a high protein score (e.g., Mascot ≥100); and (iii) a critical minimal number of peptide matches (i.e., ≥2 but usually ≥3; [103,104]). In addition, a substantial fold change (e.g., ≥2; [88,104]) is also often used as a criterion for the selection of significant ‘hits’, thus generating high-confidence proteoform identifications. In contrast, bottom-up peptide analyses generally use less stringent sequence coverage, and identification is only by inference to amino acid sequences in the available databases. Nonetheless, less stringent criteria could also be applied to increase the number of hits in top-down studies but, as with transcriptomics, the issue becomes one of establishing rigorous criteria to separate the wheat from the chaff [135]. Therefore, while less stringent criteria yield a larger number of potential canonical protein hits, from the perspective of analytical rigour, higher confidence is considered the better choice [110,113]. 

Proteome analyses thus face a number of challenges, not the least of which is sample complexity. For example, blood contains plasma, different types of cells, and thus a large number of proteins across a wide dynamic range [178]. Moreover, some canonical proteins (and all their associated proteoforms), such as albumin and immunoglobulin, are found in high concentrations, constituting ~75% of the total protein pool in plasma/serum, hampering the detection of lower abundance components [109]. To address this issue, refined methodologies or alternate biological samples can be used. For serum, the use of high-throughput gradient 2DE (i.e., multiple parallel technical replicates), utilizing a combination of two detergents (lithium dodecyl sulphate and sodium dodecyl sulphate), has been shown to improve resolution and detection of proteoforms [179]. Moreover, the sensitivity of a refined CBB staining technique, used to detect resolved protein species following 2DE, provides sensitivity to the sub-femtomole range [180]. Although sample prefractionation by affinity chromatography is commonly used to remove high abundance proteins, the resulting nonspecific loss of species cannot be ignored if genuinely quantitative analyses are the aim, as in biomarker discovery [179]. Alternatively, third separation and deep imaging can be used to complement 2DE/LC-TMS, to detect lower abundance proteins, while ensuring deeper, quantitative top-down proteome analyses which respect the native complexity of the sample (i.e., avoiding the nonspecific losses that occur during pre-fractionation steps such as affinity chromatography) [179,181,182,183,184]. 

### 5.2. Sample Handling

During proteomic analyses, appropriate procedures (e.g., sample storage conditions, use of appropriate vials, and limiting freeze/thaw cycles) are required to avoid false positive or negative outcomes. For example, one study identified the cleavage product (12.5kD) of full-length cystatin C (13.4kD) as a potential CSF biomarker of MS [185]. However, an independent follow-up study showed that the cleavage product was formed by degradation of the first eight N-terminal residues and was attributable to the inappropriate storage of samples (i.e., at −20 °C rather than −80 °C; [186]). Most importantly, the use of broad-spectrum inhibitor cocktails (e.g., containing protease, kinase, and phosphatase inhibitors) is essential to achieve high-confidence analyses that reflect the native proteome at the time of sampling [104,135,187]. Proteolysis in vitro is a nonspecific alteration of the native proteome being analysed [188] and the addition of broad-spectrum protease inhibitors is thus essential to block the degradation of proteoforms [189]. The current review found that ~50% of the CPZ, EAE, or MS studies included this critical information; others clearly did not use (or did not indicate a use of) protease inhibitors at any stage of analysis. For example, four CPZ studies reported the use of protease inhibitors [101,102,103,104], but no information of protease or kinase/phosphatase inhibitors was found in one study [105]. Many EAE studies reported the use of protease inhibitors or referred to published studies in which inhibitors were used [58,86,87,88,89,90,91,92,93,94,95,96,97,98,100]; although no information of protease inhibitors was found in one study [99]. Among MS studies, details or references to earlier protocols concerning the use of protease inhibitors was provided in some [58,62,73,76,78], but inhibitors were apparently not used in others (or at least not mentioned/found) [57,59,60,61,63,64,65,66,67,68,69,70,71,72,74,75,77,79,80,81,82,83,84,85]. We recommend a clear description of the use of appropriate inhibitor cocktails in all studies to avoid concerns regarding data quality, reliability, and reproducibility. 

### 5.3. Data Acquisition and Analysis

Protein identification using LC-TMS analysis depends mainly upon unique peptide matching, protein identification score, and the extent of sequence coverage of canonical species found in the international protein databases (e.g., Swiss-Prot; [190,191]). Notably, ‘unique’ peptides are defined as those found to correspond to one protein entry at the time of database interrogation [192]; their quality as a criterion may thus vary with time as new annotations appear in the relevant databases. Notably then, depending on the stringency of the criteria applied (which should be clearly stated in the study), canonical protein identification can thus sometimes be based simply on a single peptide (so-called ‘one-hit-wonders’; [110,193]). Protein identification score (e.g., Mascot score) relies on the observed mass spectrum that matches the stated peptides (peptide masses and peptide fragment ion masses). High peptide matching leads to higher coverage (percentage of the peptide sequence of a protein) and more confidence in the identification [190,191,194,195]. Therefore, a high identification score, in addition to the theoretical vs experimentally observed MW and pI, are the hallmarks of proteoform identification, and for some PTM this can also be complemented by selective staining following 2DE [104,123,124,196,197,198]. Sufficiently detailed unbiased analysis of peptides can also more broadly identify PTM—as opposed to affinity extractions targeting specific modifications (e.g., phosphopeptides)—and this is an important area of database development [199]. 

This review established that few studies reported all relevant information that would have been available and expected to enable critical evaluation of the data (i.e., coverage, unique peptides, score, MW, and pI; Appendix A). For example, out of five CPZ studies, only two described all the criteria [103,104], whereas score, coverage and theoretical MW and pI of the identified proteins were provided in one study [102] and only unique peptides were reported in other studies [101,105]. Of the EAE studies, tabulation of the number of peptides, score, and coverage was not found or shown in six [86,88,91,92,95,98], whereas only the number of peptide matches (i.e., no information on coverage or score) was found in four others [87,97,99,100]. Coverage and peptides were found in one study [94], and score and coverage were observed in two others [89,90]. A similar trend was found in reporting MW and pI in EAE. Both theoretical (i.e., canonical) and experimentally determined MW and pI were found in two studies [89,90], whereas no information of theoretical MW and pI was found in 10 others [86,87,91,92,93,95,97,98,99,100]. Likewise, of all the MS studies, only one described all relevant parameters, e.g., theoretical and experimental MW and pI, coverage, and score (e.g., theoretical and experimental MW and pI, coverage and score; [75]) and five studies showed data on score, coverage and unique peptides [58,62,63,79,80]. Only unique peptides were provided in three studies [77,81,83] whereas unique peptides and coverage were provided in one study [76]. Tabulation of the information of the number of unique peptide matches, score, or coverage of each identified canonical protein was not found in many studies (i.e., differentially abundant proteins were characterized based on two or more unique peptides and/or *p*-value ≤ 0.05) [59,60,65,67,69,70,72,73,74,78,82,84].

Quantification and reporting of experimental MW and pI are important as definitive indicators of PTM since most induce positional shifts in the final 2D gel proteome ‘map’ [200,201]. Moreover, doubling or tripling of MW can indicate homo-oligomerization, as shown previously in 2DE gels following CPZ-feeding [104]. This review also revealed a number of likely oligomerizations in comparing theoretical to experimental MW values including alpha-1-antitrypsin [47 vs. 105; 64], immunoglobulin kappa chain C region [11.7 vs. 28; 75], immunoglobulin lambda chain C region [11.4 vs. 35; 75], and transthyretin [16 vs. 34; 75] in MS CSF sample (Appendix A). Such oligomer formation and cytotoxicity of alpha-1-antitrypsin [202], immunoglobulin [203,204] or transthyretin [205,206] have been reported multiple times in the literature, associated with other disorders. For example, the nucleation-dependent polymerization process leads to low molecular weight transthyretin oligomer formation, which interferes with calcium influx [205]. Thus, oligomer formation may well occur in MS, but requires an appropriate analytical tool for detection; the studies to date indicate that 2DE can be one such tool. Unfortunately, some studies did not discuss potential oligomerization, suggesting that the depth of data available from 2D gels is not always fully appreciated. Oligomerization of proteins can also lead to aggregation (i.e., accumulation and clumping together; [207,208]) resulting in the subsequent formation of larger polymer ‘fibrils’ [209], although the reason why proteins become aggregated is not clearly understood [210]. Thus, without the quantitative assessment of MW, critical information is lost. The lack of such critical information in so many studies raises concerns as to the quality of the resulting protein identifications. Thus, for the purpose of this review, only if a protein from one of these particular studies had also been identified in another study using the stringent criteria necessary to fully confirm identification was that canonical protein further considered (in total 2816 proteins from MS, EAE, and CPZ; full list has been provided in Appendix A). 

### 5.4. Age Effect

MS is a disease of young people; patients can be symptomatic as early as 16 years of age [211]. Therefore, identification of protein changes in or immediately preceding the earliest phases of MS could yield much needed early biomarkers as well as enable better monitoring of disease progression. This review found that MS studies used samples from different ages such as 14–82 [78] or 24–51 [77], but no longitudinal studies for a targeted validation of proteins (i.e., changes in abundance over time) were found (e.g., sampling at 20, 40, 60 years of age). Such longitudinal studies, when feasible, are important to characterize the age-related protein changes in MS vs. normal aging. Studies with animal models also showed similar trends. For example, at study initiation, the rodents were mostly 5–8-week-old mice in CPZ [101,102,103,104,105] and 4–10-week-old mice or rats in EAE studies [87,89,90,91,92,93,94,97,98,100]. Depending upon the duration of CPZ-feeding (5–6 weeks) or EAE induction (3–4 weeks after immunization) the animals are considered young adults at the time of analysis, and this equates to ~25 years of human age [212], suggesting that proteomic results from animal studies may to some extent reflect outcomes for young adult MS patients. However, in several EAE studies, information on the age of the animals was not found [86,88,95,96,99]. A single study found proteomic changes at two different time points (5 and 12 weeks) in the CPZ model, identifying many proteoforms (e.g., ATP synthase-α, NAD-dependent protein deacetylase sirtuin-2) that changed in abundance following 0.1% CPZ-feeding [104]. However, many proteoforms that had changed in abundance relative to control animals (e.g., aconitate hydratase, rab GDP dissociation inhibitor-β) returned to control values after 12 weeks [104], suggesting that age (and adaptation) can also affect proteoform levels. Unfortunately, no longitudinal studies have been independently performed by other research groups to investigate potential age-related proteomic responses in CPZ or EAE models. Yet such longitudinal studies can identify early indicators of disease as seen in a key study with military personnel [51]. This suggests that proteomic investigations in the preclinical and acute stages are important for early diagnosis, perhaps initially in suspected cohorts (e.g., familial cases, previous severe infections). A longitudinal study (1940–1988) on patients with Epstein–Barr virus-related mononucleosis established an increased risk of MS ~30 years after infection [213]. This is also supported by a recent study in which 901 patients with early clinically isolated syndromes (i.e., within 6 months of disease diagnosis) and RRMS (within 2 years of diagnosis) proved 100% Epstein–Barr virus seropositive [214]. This suggests the crucial detrimental role of such severe early infections in the etiology of MS. Likewise, a study of neurologically asymptomatic first-degree relatives (i.e., parent, full sibling or child) of MS patients found that ~10% had T2-weighted hyper-intense lesions in the cerebrum [215]. These results suggest that MS may be comparable to (neuro)degenerative diseases, which start long before the first appearance of clinical symptoms [216,217,218]. 

### 5.5. Sex Effect

Do hormones affect protein changes in MS since the prevalence of MS (and other neurodegenerative disease) is higher in females than males [219]? Interestingly, sex (hormones) plays an important role in BBB disruption reviewed in [220]. This review found 19 MS studies that used samples from both sexes [58,60,61,62,63,65,67,68,69,70,71,72,73,75,79,80,82,83,84], while seven studies used only women [57,64,66,76,77,78,81]. Unfortunately, no information about sex was found in one study [59]. However, a focused comparative study investigating differential sex-dependent regulation (i.e., comparing protein changes between men and women) remains to be investigated. Similar data were found in animal studies. For example, a similar number of studies on males [104,105] and females [101,102,103] were found using CPZ. On the contrary, nine EAE studies used females [87,89,90,91,94,97,98,100] but only two used males [92,99], and no information on sex was found in several studies [86,88,95,96]. Is there any sex difference in terms of the CPZ or EAE models? Histological findings indicated no sex differences in the CPZ model [221], but clear differences in EAE. For example, female EAE mice show greater demyelination and rapid onset of disease development compared to males [222]. Likewise, in EAE, neurological deficits (e.g., neuropathic pain) are greater in females relative to male mice [223]. However, neither the EAE nor CPZ animal model has been used to investigate the sex-specific differential regulation of proteins and their involvement in BBB disruption. 

## 6. Discrepancies between Animal Models and MS at the Proteome Level

Although no animal models faithfully mimic the complete complexity of MS, the appropriate use of models depends upon the specific research question being posed [12]. Despite the differences in disease induction in EAE (peripheral injection of myelin protein) and CPZ (feeding of toxic compound), both experimental models show oligodendrocyte degeneration, demyelination, and glial activation in the CNS [14,224]. In EAE, the CNS (mainly spinal cord) is infiltrated by adaptive immune cells whereas no such cells are found in the CPZ model without select modifications to the standard protocol [9,10]. Comparative studies could explain the magnitude of similarities and differences between animal models and MS itself, at the proteome level, providing new directions to develop models that better reflect the complexity of MS (i.e., both symptomology and pathophysiology). This review thus investigated the currently published evidence as to how effectively animal models mimic human MS, by comparing changes in the abundance of protein species identified in MS samples with those identified in CPZ and EAE (i.e., either an increase or decrease irrespective of the sample type analysed). Five CPZ studies reported 155 canonical proteins that changed in abundance (Appendix A); of these, 91 showed an increase in abundance whereas 64 decreased [101,102,103,104,105]. In contrast, fifteen EAE studies identified 2139 canonical proteins that changed in abundance; 974 increased and 1165 decreased (Appendix A). MS studies identified abundance changes in 523 proteins across 28 studies; 340 increased in abundance and 183 decreased (Appendix A). Table 3 summarises the protein changes found to be common among MS, EAE, and CPZ, irrespective of the type of sample analysed (i.e., both intra- and inter-sample comparisons). Only a few proteins that changed in abundance were common between CPZ and MS, whereas a greater overlap was found between EAE and MS (Table 3). Overall, however, these represent a rather low number of similarities out of over 2000 identified protein changes (Appendix A) indicating that neither EAE nor CPZ mimic clinic MS particularly well at the proteome level. Moreover, assessment of the known functions (e.g., metabolic, blood-related, immune response) of these common proteins (Table 3) revealed marked similarities between EAE and MS (53%), but less similarity between CPZ and MS (only 2%, Appendix A). It is, however, noteworthy that most of the EAE studies used CNS samples (mainly spinal cord), but MS proteomic research was mainly focused on CSF; however, the apparent similarity between EAE and MS may indicate some similar pathology since CSF contains mainly CNS proteins [52,153,154]. Lists of all samples used in all the reviewed studies are detailed in Appendix A. It is also noteworthy that the total number of EAE studies in the literature is substantially higher than those using CPZ (~13,659 vs. 888, PubMed as of June 2020). In MS and EAE, a large number of changes in protein abundance were identified in blood and this was attributed to the disruption of the BBB, whereas in CPZ no such proteins were found (Appendix A). Moreover, in MS and EAE, proteins related to the adaptive immune responses (e.g., immunoglobulin and complement) were more common, whereas proteins of the innate immune response were identified in CPZ (e.g., glial fibrillary acidic protein, GFAP; Appendix A). Due to these immunological similarities between EAE and MS, EAE has been widely considered as ‘the’ model of MS [12,225]. However, samples from MS patients were collected after clinical diagnosis or post-mortem [58,76]. Protein changes identified in post-mortem samples were heavily skewed toward involvement in immune response and did not provide information about the pathoetiology of the disease, but only about its later and longer-term, progressive consequences.

In MS and its animal models, histological examinations indicate a reduction in myelin proteins [5,226,227]. However, proteomic analyses did not replicate the histological findings and changes in myelin proteins were only identified in a few MS studies, but with contradictory results [58,76]. For example, the abundance of cerebrum myelin basic protein increased in one study [58] but was reduced in another [76]. In CPZ studies, myelin basic protein, myelin proteolipid protein, and myelin-associated glycoprotein levels were reduced [105]. In EAE, changes in the abundance of myelin basic protein were reported in three studies in both the cerebrum and spinal cord, but the results were contradictory; two studies found a decrease in abundance [87,92], but the other identified an increased abundance [58]. These conflicting results in EAE may depend upon the use of different exogenous CNS myelin antigens (e.g., myelin basic protein vs myelin oligodendrocyte glycoprotein) and/or the animal strain (e.g., C57Bl/6 vs. SJL/J); based on such differences, EAE has been said to mimic either relapsing-remitting or the progressive form of MS [224]. Fifty percent of studies used myelin oligodendrocyte glycoprotein to induce EAE [87,89,90,91,92,95,96,98], while 38% used myelin basic protein [88,93,94,97,99,100], and 12% used proteolipid protein [86,95]. Whether the contradictory results concerning protein alterations arise from different antigens or strains used remains unclear. However, Jastorff and co-workers performed a comparative analysis, inducing EAE in SJL/J mice using two myelin antigens (myelin oligodendrocyte glycoprotein and proteolipid protein). This study showed only a subtle difference in spinal cord proteomes (i.e., only a few differential protein changes, such as cofilin either increasing or decreasing in abundance) [95]. A similar contradiction in proteomic versus histological data was observed for the astrocyte protein GFAP, a common marker for this cell type [228,229]. Histological analysis showed that expression of GFAP positive astrocytes increased in post-mortem MS CNS samples [5,230] but this was confirmed in only one proteomic study [76]. Multiple proteomic studies of the CPZ cerebrum [104,105] and EAE brain stem and spinal cord also reported an increased abundance of GFAP [88,90,92,96,100]. Thus, while different methods have their own advantages and limitations, these observations collectively indicate that data from animal studies (EAE and CPZ) simply do not consistently mimic the protein changes seen in human MS. 

Although ~80% of lab-based biomedical studies are performed using mice or rats, it is now more widely accepted that animal studies do not fully (or necessarily effectively) mimic human disease phenotypes [212]. Therefore, data from animal models must be interpreted with caution and via the most thorough possible assessments relative to the human condition. At the simplest level, MS may not be mimicked in mice or any other rodents for a number of reasons ranging from the fact that they are not bipedal to the differences in their immune systems. For example, neutrophils are the predominant cells in human blood (50–70% neutrophils, 30–50% lymphocytes) whereas mice have a strong preponderance of lymphocytes (75–90% lymphocytes, 10–25% neutrophils) [231]. Humans are also ~2500 times larger than mice or rats and their lifespan is ~20 times longer [212,232]. These variations amount to huge differences between mice and humans that are not only limited to metabolism, or immune function, but also environmental conditions, including specific temperature and humidity, as well as diet (e.g., lab rodents are generally fed with only autoclaved and thus pathogen-free chow). In short, there is no significant diversity between test subjects as there is with human patients (e.g., in terms of lifestyle, including different living conditions eating habits, sleep-wake cycles, and so forth) and thus results from mice or other rodents often do not closely replicate what is experienced by humans [212,233,234]. Furthermore, variations in housing conditions and the nature of handling (e.g., the gender of the handler) can produce marked inter-lab variations in results obtained from rodents [235,236,237]. 

## 7. Differences among MS Phenotypes at the Proteome Level

Protein changes across the different phenotypes of MS were also investigated, and some were found to be consistent while others differed (Appendix A and Table 4). For example, fibrinogen was significantly increased in abundance in the blood of PPMS patients compared to those with RRMS [73]. Only a few proteins showed comparable changes between phenotypes. For example, an increase in the abundance of beta-2-microglobulin was observed in CSF of PPMS, SPMS, and RRMS samples [64,65,75,82]. These differences in protein profiles may be due to the nature of disease initiation (e.g., viral infection, oligodendrocyte degeneration, or autoimmunity; [238,239]). Moreover, age and the localization of demyelinated lesions can affect disease outcome. For example, PPMS differs from RRMS or SPMS in terms of disease presentation; PPMS symptoms appear later in life and affect men and women equally with a predominance of lesions in the spinal cord rather than the brain [238]. However, it is clear that the number of CNS proteins changing in abundance is higher in RRMS compared to other MS phenotypes (Table 4 and Appendix A). It is thus noteworthy that over 50% of proteomic studies used RRMS patient samples, since this is the most common form of MS [14,15]. Interestingly, a large number of identified proteins were also found to be involved in the pathophysiology of other neurodegenerative conditions such as Alzheimer’s and Parkinson’s diseases (see below). This suggests that MS may also have an underlying slow, degenerative nature and that the hallmark clinical signs and symptoms appear only long after disease initiation [216,217,218]. For example, a landmark longitudinal study [51] found evidence of axonal degeneration as early as 6 years prior to the onset of clinical MS symptoms and MRI-detectable lesions using serum samples collected from US military personnel from 2000–2011 and analysed in 2018–2019 using a single-molecule array assay [51]. However, it is acknowledged that not all the patients with radiologically isolated syndrome progress to MS. Nonetheless, ~2–5 years after diagnosis of radiologically isolated syndrome (using MRI-identified lesions) ~30% of patients convert to a clinical diagnosis of MS [22,240]. The finding of elevated levels of serum neurofilament light chain (an axonal protein marker; [162]) in the presymptomatic stage thus indicates that MS may well have a prodromal phase lasting several years [51]. Furthermore, analysis of blood samples from pediatric MS patients detected six common proteins (across different MS phenotypes; Table 4)—alpha-1-acid glycoprotein, alpha-1B glycoprotein, apolipoprotein, clusterin, gelsolin, and vitamin D binding protein [79]—suggesting that some alterations of key proteins occur early in life and are subsequently sustained. Together, these studies highlight the importance of early routine testing of a readily available sample (e.g., blood) for select biomarkers, perhaps during adolescence (i.e., a presymptomatic stage of MS).

## 8. Differentially Abundant Canonical Proteins in MS and Animal Models

Table 3 and Table 4, and Appendix A summarise the differentially abundant proteins identified in MS, EAE, and CPZ publications. They were classified into functional categories using detailed literature searches (Appendix A). The majority of these proteins are related to blood, metabolism, immune response, the proteasome, aggregation, and myelin, indicating that protein changes in MS and animal models are not linked with only one biological process or functionality, but rather are consistent with the multifactorial nature of MS [241,242]. However, this comparison (intra- and inter-sample) was based on the limited published studies from MS-like animal models (EAE-15 and CPZ-5 studies) and MS itself (29 studies); one must also be cognizant of the likelihood of file-drawer effects being at play as well (i.e., publication bias if results do not support favoured hypothesis). Clearly, more studies are required to better understand changes in proteoform abundance (i.e., increase or decrease) in MS. These studies should include multiple sample types (e.g., blood, urine, tear) from different animal models (e.g., EAE, CPZ) and MS patients assessed in parallel to investigate whether the same proteoform changes in a given sample from MS patients are also found in the same samples from animal models. These studies would directly establish the effectiveness of animal models currently used to investigate the pathoetiology of MS and identify critical biomarkers and, potentially, new therapeutic targets enabling the most effective early intervention. 

### 8.1. Blood-Related Proteins

Table 4 shows that at least 16 blood proteins were altered in abundance in all forms of MS. Moreover, the data in Table 3 and Appendix A indicate similar alterations in EAE. Conversely, these proteins were not detected in any CPZ studies, consistent with the preserved integrity of the BBB [104,243]. Notably, abundance changes in some proteins, such as albumin and vitamin-D binding protein were identified in most MS phenotypes. The presence of albumin in the CSF samples of MS patients has been reported in multiple studies, but with contradictory results indicating either an increase [61,63,64,68,80] or decrease [70,75,84] in abundance. Similar observations were made in EAE studies; five out of six studies showed an increased albumin abundance in cerebrum and spinal cord [87,88,96,97,100] whereas the remaining study identified a decrease in the cerebrum [92]. A change in the abundance of vitamin D-binding protein was observed in several MS-related studies, either as an increase [60,63,64,79] or decrease [70,75,82] in both CSF [60,63,64,70,75,82] and blood [79]. Does the differential outcome with regard to vitamin D-binding protein depend upon the sample or method used? Three out of four studies that found an increase correlated with use of a top-down approach to analyse CSF [63,64] and plasma [79]. In comparison, both top-down [75] and bottom-up [70,82] methods identified a decrease in the abundance of vitamin D-binding protein in CSF. For top-down analyses, silver was used to stain gels for 20 minutes [63] and CBB was used for 12 [64] or 16 hours [75]. We found no evidence indicating that incubation time or staining method effects the detection of vitamin D-binding protein. Based on recent work in preterm labor it seems likely that specific proteoforms of vitamin D-binding protein are altered in abundance and that with multiple potential proteoforms, abundance data may be skewed one way or the other depending on the analysis used and the quality of the data obtained [244]. 

An inconsistency in the protein changes between the periphery and CNS may also reflect efflux limitations (even with a compromised BBB), and/or further processing of proteoforms (in or outside the brain) that then may appear to change in abundance in a manner opposite to that of the initial species. Nonetheless, such changes with disease do not indicate that a peripheral measure is less useful as a biomarker, as a validated biomarker need not be directly related to the disease mechanism. Moreover, no reliable trend (either increase or decrease) was identified among proteins or studies. This inconsistency in results included contradictory changes in abundance of a protein (either increasing or decreasing) depending on the method (e.g., top-down or bottom-up) used, even when the same sample type (e.g., CSF) was analysed from the same model (e.g., EAE) or from MS patients. Therefore, more research is required to fully understand how blood-related proteins contribute to MS pathology; targeted proteomic analyses (e.g., Western blotting for specific proteoforms) are required to effectively validate the ‘proteins’ identified to date as potential MS biomarkers. 

### 8.2. Metabolism 

A large number of metabolic proteins such as apolipoproteins and mitochondrial proteins were found to be altered in abundance in different samples from MS patients, including CSF [59,60,63,64,66,69,70,71,72,75,77,78,82,83], cerebrum [58,76], tears [80], urine [81] and blood [57,62,79,85]. In MS, an increased abundance of apolipoproteins (A, C, D, and E) were reported in several studies of CSF [59,60,63,64], blood [79] or tear samples [80], but a decreased abundance (of apolipoproteins A, D, and E) has also been reported in CSF [70,71,72,75] and tear samples [80]. Notably, apolipoproteins AI, AII, and D have been reported in tear samples, with AI and AII increased in abundance but apolipoprotein D decreased [80]. Does the increase or decrease in the abundance of apolipoprotein depend upon the analytical method used? Apolipoprotein E has been identified in six different top-down studies (using CSF samples), with an equal number reporting an increase [59,63,64] or decrease [71,72,75] in abundance. No bottom-up study identified apolipoprotein E. Of several studies [59,60,63,64,69,70,71,72,75,79,80], validation of the abundance of apolipoprotein E using Western blot analysis was only found in one [71]. Overall, contradictory results across different studies reduce the usefulness of apolipoprotein as a potential marker for MS, unless of course specific proteoforms can be conclusively shown to selectively change in abundance. Moreover, changes in the abundance of apolipoproteins are also associated with several chronic diseases such as Alzheimer’s [245,246] and Parkinson’s [247] diseases.

As shown in Table 3 and Table 4, and Appendix A, numerous studies also identified changes in antioxidants in MS, CPZ and EAE. In MS, glutathione peroxidase in CSF [63] and peroxiredoxin-6 in the cerebrum [76] were identified. Increased abundance of peroxiredoxin-2 and peroxiredoxin-6 was detected in the cerebrum of CPZ mice [105]. In EAE, changes in the abundance of peroxiredoxin-1,2,4,5 and 6 were detected in the cerebrum [95], brain stem [100], and spinal cord [89,92,94], but with contradictory results both in top-down [89,95,100] and bottom-up [92,94] studies. These differences in the trends of protein abundance may in part be due to the use of different analytical techniques. For instance, when CyDye was used as a pre-electrophoretic protein label, a reduction in the abundance of these proteins was detected [100]; in contrast, an increase in the abundance of these proteins was indicated when either silver [89] or CBB [95] were used to identify the native species. Whether the contradictory outcomes regarding peroxiredoxin protein were due to different staining approaches (e.g., CyDye vs silver or CBB), pre-labelling incubations, sample handling or preparation, or the spot identification and excision process, remain untested. For example, use of CyDyes for protein labelling is compromised by several factors such as limited resolution at higher pI values, poor separation when proteins are highly acidic and basic, and protein location shift due to labelling [248]. Whether any of these factors effected the study outcomes regarding this protein (and others) remains untested. 

Do metabolic protein changes in MS and animal models link with oligodendrocyte degeneration? Literature mining revealed that most identified canonical proteins were metabolic (i.e., EAE (25%), CPZ (37%), and MS (24%); Appendix A). This strongly supports the previous proposal that metabolic dysregulation leads to the degeneration of oligodendrocytes [10,104]. Importantly, these metabolic changes are detected in CPZ-fed mice prior to the detection of any overt demyelination by conventional tools, consistent with biochemical changes preceding oligodendrocyte degeneration and subsequent demyelination [13]. However, most proteins (~80%) do not exert their biological functions alone but rather work with other proteins [249,250]. Yet, assessment of the available data identified no category-related interactions (e.g., metabolic) among commonly identified canonical proteins (Table 3 and Appendix A), using the STRING program (https://string-db.org/; accessed in May 2020) for protein–protein interaction analysis [104,251]. However, there are clear examples of some of these being proteoforms of the canonical sequences (Appendix A). In contrast, by using the PANTHER program (http://pantherdb.org/; accessed in May 2020), a mixed complex interaction and association of many pathways was identified (Appendix A). These are characterized by, for example, combinations of metabolic, immune-, blood-, and myelin-related proteins, consistent with the multifactorial nature of MS initiation and progression.

### 8.3. Immune Response

Many immune system-regulating proteins such as complement, immunoglobulin, and GFAP proteins were detected in samples from MS patients as well as animal models (summarised in Appendix A). Interestingly, the differential abundance (i.e., increase or decrease) of the same proteins was found using both top-down [63,64,72,79,80] and bottom-up [69,70,84] techniques. This inconsistency in results across studies was observed for proteins even with the same sample type from the same phenotype of MS (e.g., RRMS). For example, complement factor H was found to either increase [75] or decrease [72] in abundance when CSF samples from RRMS patients were assessed using the same method (i.e., two-dimensional difference gel electrophoresis). Complement C3 abundance increased in CSF samples assessed using the top-down approach, whereas complement factor H decreased in the same sample [72], suggesting differential regulation of complement protein subtypes, which may contribute to selective activation of complement pathways (e.g., classical, lectin or alternative) [252]. In contrast, a consistent increase in the abundance of complement C3, C4, H, and I in CSF samples was detected in another top-down study [63]. Notably, the latter used pre-electrophoretic reactive labelling with CyDyes while another stained with silver after the resolution of proteoforms [72]. Notably, complement proteins (complements C3, B, and I) were also identified in blood [79,84] and tears [80], suggesting that tears may be an effective alternative to sampling CSF. However, a localized difference in complement proteins was observed in the EAE model; a bottom-up assessment identified an increased abundance of these proteins in the cerebrum [92], cerebellum [92], brain stem [92], spinal cord [87,92,97] and CSF [99], but the abundance of complement component 1Q subcomponent-binding protein was reduced in the spinal cord [92]. Notably, none of the EAE studies that used a top-down approach detected complement proteins. However, a top-down assessment of the CPZ model revealed a reduced abundance of one complement protein (C1qbp) in peripheral blood mononuclear cells [103]. The reduced abundance of complement complex following CPZ ingestion reflects the general suppression of peripheral immune responses as reported in other studies [9,11,104,253,254]. Not surprisingly, complement protein activation is not unique to MS or its animal models and is found in other neurological diseases and non-neurological disorders [183,255,256]; thus, it seems unlikely that these could be used as sole, definitive biomarkers of MS. Several studies identified changes in immunoglobulins (e.g., alpha, gamma, kappa, and lambda) in tears [80], urine [81], blood [57], and CSF [59,60,61,63,64,70,71,75,83], but again with no consistent trend. Although immunoglobulins are already used as general biomarkers for MS diagnosis [6,35,36], it is worth considering their limitations: first, like the other protein functional groups discussed above, there is no consistent trend of changes in immunoglobulin abundance in MS. Thus, while several MS studies reported increased immunoglobulin abundance [59,60,61,63,64,70,75,80,83], others found the opposite [57,71,75,81]. There is also a clear dependence on the type of sample analysed. Perhaps a full comparative screen of immunoglobulins in 2–3 sample types (e.g., CSF, blood, and urine) from MS patients and EAE and CPZ models would help to clarify whether or not any of the immunoglobulins might serve as more selective MS biomarkers.

Changes in the abundance of innate immune response-regulating proteins have also been reported in MS. For example, S100, a calcium-binding cytosolic protein that plays a significant role in regulating macrophage-mediated inflammation [257], was identified in blood [57,85] and tear [80] samples from MS patients. Similarly, increases in the abundance of neuroinflammatory proteins GFAP [229] and vimentin [258] were reported in MS patient cerebrum [76] and CSF [77], respectively. 

Overall, the literature regularly reported increased or decreased abundance of immune-related proteins, irrespective of whether the same or different samples were used, or whether they were from the same MS phenotypes or animal models. These variations depended, to some extent, upon the methodology used (i.e., top-down or bottom-up) or the protein detection method (e.g., CyDye labelling or gel staining with silver or CBB). In addition, proteins identified in these studies of MS also change in other neurological diseases. Thus, to date, it would seem that a reliable biomarker for MS remains undefined. 

### 8.4. Structural Changes 

In MS and animal models, marked changes were observed in structural proteins including actin, coronin, gelsolin, fibulin, vinculin, integrin-β, β-fibrin, and tubulin (fully summarised in Appendix A). These changes are perhaps not surprising since cytoskeletal proteins are critical for the neuro-architectural organization and are involved in stress responses [259,260]. However, again, reports in the literature are contradictory. For example, in MS, actin proteoform abundance in CSF samples from MS patients was reported to increase in two studies [62,63,64] and decrease in another [75]. Likewise, the abundance of the actin-binding protein, gelsolin [261], was found to increase in three studies using CSF or blood samples [63,64,79] yet decreased in two others using CSF [70,72]. However, changes in structural proteins including actin and tubulin are a common phenomenon in neurodegenerative diseases including Alzheimer’s, Parkinson’s, and amyotrophic lateral sclerosis [262,263,264]. Changes in many structural proteins were also reported in the EAE and CPZ models (Table 3 and Appendix A), suggesting that structural alterations also underlie neurodegeneration in MS patients [218,265]. 

### 8.5. Proteases and Protease Inhibitors

Changes in several proteases and protease inhibitors have been reported. An increase in the abundance of vitronectin, a multifunctional protein which plays a role in the regulation of cell adhesion and invasion [266], was observed in two independent studies in CSF samples from MS patients [60,64]. In contrast, kallikrein 6 abundance either increased [63,64] or decreased [65,69,70,75] in CSF from MS patients. In EAE, only one study showed an increase in the abundance of kallikrein 6 in spinal cord samples [92], whereas, in the CPZ model, no changes in kallikrein 6 have been reported. In addition to proteases, changes in the abundance of many protease inhibitors (e.g., cystatin and antichymotrypsin) have been reported, mainly in MS and EAE but, again, with inconsistencies between studies. For example, the abundance of antichymotrypsin, which plays an inflammatory role [267], was found to increase in six studies [60,64,66,67,78,80] but to decrease in another [69]. These studies used different samples and methods. Notably, all top-down studies identified a consistent increase in the abundance of antichymotrypsin in CSF or tears [64,66,78,80], whereas bottom-up studies identified increases [60,67] or decreases [69]. Likewise, five studies that detected changes in antitrypsin found either increases [63,64,84,85] or decreases [75] in abundance even when the same sample type and method was used. However, these top-down CSF analyses used different pH ranges for isoelectric focusing, stains (i.e., CBB or silver), as well as different staining times—strongly suggesting that further investigation is required to establish the actual status of antitrypsin in MS. A reduction in the abundance of serine protease inhibitor has also been reported in CSF [71]; this may be critical as such enzymes may be directly related to the apoptotic and neurodegenerative processes [268,269]. However, changes in the abundance of proteases and protease inhibitors are not unique to MS but are also seen in other neurodegenerative diseases [267,270]. Changes in the abundance of proteases may explain why inflammatory responses are high and increase with the progression of MS. Moreover, the increased abundance of proteases may trigger alterations in molecular chaperones, a ubiquitous family of enzymes that regulate protein folding and assembly; the dysregulation of molecular chaperones may lead to protein aggregation [271,272,273]. 

### 8.6. Protein Aggregation

A number of molecular chaperones including clusterin, heat shock protein, and calnexin were reported in multiple proteomic studies (Table 3 and Appendix A). Increased abundance of heat shock protein, which is activated in response to stressful stimuli and is important for protein folding dynamics [274], was observed in tears and peripheral blood mononuclear cells from MS patients [62,80]. Moreover, changes in clusterin were reported in five independent studies in which its abundance was found to either increase [64,69,79] or decrease [70,75], seemingly with some dependence on the analytical method used. Changes in the abundance of molecular chaperones were also identified in the EAE [86,87,88,89,90,92,94,95,100] and CPZ models [101,103,104,105]. Moreover, we [104] and others [64,71,75] found evidence of protein self-association or oligomer formation [104,275]. Notably, polymerization and aggregation are fundamentally comparable, aggregation referring to a nonspecific process and polymerization to a more defined one [210]. Evidence of protein aggregation in MS samples was shown in another study that described the presence of soluble oligomers (without specifying the protein(s) involved) in the cerebrum and CSF of MS patients [208]. This suggests that proteins were misfolded and eventually aggregated [208,276]. Such protein interactions can enhance oligodendrocyte degeneration that subsequently results in demyelination [277,278,279]. For example, endoplasmic reticulum-induced stress response (measured by overexpression of the folding enzyme disulphide isomerase) leads to apoptosis of oligodendrocytes [280]. This review found that changes in protein disulphide isomerase were reported in MS blood samples [57,62], brain stem and spinal cord of EAE animals [88,92,96,100], and blood and spleen of CPZ-fed mice [103]. However, these reports were also inconsistent, indicating either increasing [62,88,92,96,100] or decreasing [57,103] abundance. Thus, evidence from MS samples [57,62,208,281] and animal models [88,92,96,100,103,104,282] suggest that the unfolded protein response and protein aggregation can play a role in the development and progression of MS; however, these also play a role in neurodegenerative diseases including Alzheimer’s, Parkinson’s and Prion diseases [283,284]. 

### 8.7. Demyelination and Axonal Injury

Demyelination is the pathological hallmark of MS [5,285]; therefore, changes in myelin-related proteins are to be expected. In MS studies, most of the myelin protein changes were found in CSF [60,63,64,67,69,70,78], cerebrum [58,76], urine [81], and blood [79]. Changes in the abundance of myelin component proteins have also been reported in both EAE [87,89,90,91,92,94,95,100] and CPZ [104,105] but with both increases and decreases identified. While a reduced abundance of myelin basic protein is commonly taken to indicate demyelination, some EAE studies have reported an increased abundance in the spinal cord [94], whereas decreases were identified in the whole cerebrum, cerebellum, brain stem and spinal cord in other studies [87,92]. Likewise, the abundance of both myelin-associated glycoprotein [92,94] and myelin oligodendrocyte glycoprotein [92] was reduced in the spinal cord, but the abundance of myelin proteolipid protein reportedly increased [94]. Notably, none of the top-down analyses of either EAE or CPZ (e.g., in cerebrum) detected any significant changes in myelin basic protein, myelin oligodendrocyte glycoprotein, myelin-associated glycoprotein, or myelin proteolipid protein. It may thus simply be the case that the hyper-abundance of these species vs the localized nature of the demyelination [166] results in only a very low overall change in the amounts of these proteins in whole cerebrum samples. While demyelination correlates with axonal injury in MS [286], whether axonal injury and demyelination run parallel remains unproven since overt axonal injury can also be seen in subjects with subtle demyelination [287] presymptomatic, normal-appearing white matter [20], or even independent of MRI-visible inflammation [21]. Axonal markers such as 14-3-3 protein [67,81,89,90,94,104], neurofilament [67,75,88,92,93,94,98,104] and amyloid precursor protein [64,70,79,91,92,95] were identified in both MS patients and animal models but, again, with contradictory results. For example, an increased abundance of amyloid-β was found in blood [79] and CSF [64] from MS patients, but a reduced abundance was reported in another study of CSF [70]. 

Detection of the axonal injury marker, amyloid precursor protein, in the blood of young MS patients (age ~15 years) may indicate that neuronal degeneration occurs at an early stage of disease development and/or is indicative of its progression [79]. By quantifying markers of axonal damage early in life, it is hoped that the permanent neurological impairments in MS can be avoided or at least substantially delayed and reduced [286,288]. Notably, while the presence of elevated concentrations of neurofilament light chain in blood and CSF is considered a potential monitoring biomarker of MS [162], only one proteomic study reported an elevation of CSF neurofilament light chain (and the heavy chain) in SPMS [94]. However, the most sensitive method of detecting neurofilament light chain is a targeted proteomic assay—the single molecule array—having picogram detection limits [289]. In this regard, the targeted assay is more sensitive than the discovery assay. However, recent developments (e.g., use of high sensitivity staining, third separations and deep imaging) in 2DE-based top-down discovery analyses can detect species even at the sub-femtomole level [135,179,180]; these strategies remain to be tested in terms of detecting neurofilament light chain in samples from MS patients or animal models. Nonetheless, the presence of axonal injury markers is not unique to MS and these are also found in other CNS diseases, including in the CSF of Alzheimer’s [290] and Parkinson’s [291] disease patients, further indicating that careful evaluation is a prerequisite for the diagnosis of MS. Again, detection of differences in specific proteoforms would prove most useful in differentiating between different diseases. However, despite similar changes in some canonical proteins in MS and other neurological diseases, no study was found that investigated the cut-off values, which may enable a potential differential diagnosis of MS. Likewise, no study was found that determined cut-off values for any proteins that may differentiate MS phenotypes (e.g., RRMS, SPMS). Certainly, there has been no effective assessment of potential differences in proteoforms of any of these common canonical proteins; this would provide for the most definitive differential diagnosis of MS and/or its phenotypes.

During the preparation of this manuscript, some new studies were published (i.e., 2020 to February 2021) describing proteomic changes in samples from MS patients [292,293,294] and animal models [295]. Bottom-up studies using CSF samples from MS patients reported the upregulation of apolipoproteins, augurin, receptor-type tyrosine-protein phosphatase gamma, and trypsin-1 compared to controls. Notably, samples from SPMS showed the greater number of changes in proteins (compared to PPMS, or relapse stage) [294]. This suggests a unique change in proteome profile when MS progresses with little or no inflammation in the secondary progressive form of MS [294]. Moreover, the changes in proteins involved with neural development (e.g., semaphoring-7A, neural cell adhesion molecules, transforming growth factor beta 1, follistatin-related protein 1, transferrin) were reported [292]. These changes are similar to those in the samples from MS patients and those diagnosed with clinically isolated syndrome. These findings suggest that disruption of neural structures may represent an early stage of MS development [292]. Importantly, a bottom-up study was also performed on CSF samples from newly diagnosed female RRMS patients (with mild neurological disability and no long-term drug therapy) and revealed the differential regulation of 69 canonical proteins including immunoglobulin, apolipoprotein, and complement [293]. Most of these canonical proteins had been identified in previous studies of patient samples taken at different disease stages (e.g., PPMS, SPMS), suggesting that these proteins or perhaps specific proteoforms thereof may be used to monitor MS from early clinical stages and through progression. 

Szilagyi et al. used a bottom-up (iTRAQ) approach to look for proteomic changes during the demyelinating phase in CPZ-fed mice [295]. While most previous CPZ studies used whole brain samples for proteomic investigation [102,103,104,105], Szilagyi et al. [295] used only corpus callosum, similar to an earlier study [101]. Martin et al. [101] identified changes in the abundance of 19 canonical proteins, also using iTRAQ, while this new work by Szilagyi et al. [295] identified 191 canonical proteins changing in abundance. Only one common protein (increased abundance of signal transducer and activator of transcription 1) was detected in both studies suggesting, again, a similarity with the seemingly routine inconsistencies between studies already discussed in this review (i.e., very little consistency in proteins identified across studies). Martin et al. [101] used transgenic miR-146a knockout mice, whereas Szilagyi et al. [295] used C57Bl/6 mice, suggesting that a strain-dependent difference may, at least in part, underlie the differences in the study results. 

## 9. Conclusions

This review was designed to discuss why the identification of key proteoforms is critically important to investigate MS pathoetiology and to identify potential (early) biomarkers related to demyelinating conditions like MS. However, while the extensive review and comparative analysis of the published literature revealed many potential biomarkers that reflect the multifactorial pathoetiology of MS, the majority also overlap with other neurodegenerative diseases. However, few studies actually identified proteoforms but rather canonical proteins, and fewer still reported any consistent trend (i.e., increase or decrease in abundance) either between sample types (many even in the same sample type), analytical methods (i.e., top-down vs. bottom-up, and technical variations in each of these), or studies. Thus, extensive validation studies would be necessary to rule out false-positive and false-negative results in the published data. Much of this confusion seems likely to arise from a lack of attention to the analysis of proteoforms [110,111,113,124,134,244,296], which would require a more concerted and detailed effort to identify critical protein species rather than simply cataloging canonical proteins. Certainly, there are data (Appendix A) consistently identifying at least nine canonical proteins across studies, with the top-down analyses specifically identifying these as particular proteoforms of the generic amino acid sequence (i.e., there is a notable variance between the theoretical and experimentally observed pI and/or MW). The critical question thus becomes whether specific proteoforms of some commonly identified canonical proteins can discriminate very early and progressive stages of MS, as well as MS from other neurological diseases. However, the data summarised here make it clear that the advent and extensive adoption of shotgun analysis has not promoted a deeper understanding of underlying, initiating mechanisms, nor led to the identification of definitive (early) biomarkers. Nonetheless, overall, metabolically dysregulated proteins were found as a major cluster across animal models and MS, suggesting that some as yet undefined internal metabolic dysfunction in oligodendrocytes (or other local cells such as astrocytes) may initiate MS [12,13]. Overall, we conclude that:Differential changes occur in protein abundance irrespective of whether the samples analysed are from MS patients or animal models. While many discrepancies in the findings are methods-related (top-down vs bottom-up), many differences are also sample-related, which may be due to the sample types analysed and/or to inter-lab variations in sampling, sample preparation, and sample handling.Although both methods have advantages and limitations, taking all the different analytical approaches into consideration, we recommend the use of integrative top-down over bottom-up analyses, since this is sensitive, has the highest inherent capacity to resolve intact proteoforms (i.e., quantitatively addresses the inherent complexity of proteomes), yields excellent sequence coverage, and provides a high degree of consistency across technical and biological replicates.Due to the inherent complexity of CSF collection with potential confounding blood contamination, CSF is not recommended for routine proteomic analysis; alternatively, easily accessible biological samples such as blood, tears, and urine can be used. Regrettably, these have largely not been utilized or have been handled and analysed with such a divergence of protocols, that few studies can realistically be considered as replicated.Neither the literature search nor the bioinformatic analyses revealed any single functional category of proteins but rather a range of different functionalities. Therefore, the published data reinforce the multifactorial nature of MS disease initiation and progression. Consequently, no reliable protein let alone proteoform biomarker, has been identified, and it seems likely that a panel of well-validated biomarkers will be necessary and could include specific validated proteoforms, lipids, and metabolites.Many changes in protein abundance identified in MS also occur in Alzheimer’s or Parkinson’s diseases, suggesting that critical mechanisms underlying MS may well be neurodegenerative. This also further complicates the search for effective early biomarkers specific to MS. Again, it may be that specific proteoforms are proven to have discriminatory power, but deep, quantitative proteome analyses will be needed to establish this.A common biomarker in MS and other neurodegenerative diseases could be useful for different stages of life. Detecting specific proteoform changes at younger ages may be useful to identify MS. However, further research will be necessary, as it is now apparent that conditions such as Alzheimer’s disease likely also have an earlier prodrome than their clinically identifying sequelae imply. Moreover, if we consider MS as neurodegenerative then many proteins (although perhaps not proteoforms) associated with multiple diseases will likely be proven to be common. Again, the critical question then becomes whether specific proteoforms may be selective biomarkers for the different diseases and their stages.Neither EAE nor CPZ show extensive similarities with MS, indicating that the current animal models likely only poorly mimic MS, at least at the proteome level, but may be useful if consensus in the field can be reached regarding the most likely underlying pathway to MS (i.e., inside-out or outside-in). The available animal models could then be used in more targeted studies to explore potential initiating mechanisms (i.e., CPZ) vs those underlying later stages of progression (i.e., EAE). This also emphasizes the importance of developing better animal models for MS to move beyond longstanding research approaches and viewpoints that have often become somewhat dogmatic.A lack of consistent procedures in proteomic analyses and the failure of journals to demand the necessary rigour in both methods and data reporting have yielded a literature of contradictory results. This has substantially delayed the identification of definitive proteoform biomarkers and therapeutic targets that directly underlie the fundamental molecular cause(s) of MS. Future research must thus clearly focus on the identification of consistent changes in specific proteoforms rather than canonical proteins.A broader agreement about consistency of analytical approaches is required, rather than the somewhat random choice of samples, analytical methods, and models that the breadth of available literature currently suggests is the case. Perhaps larger, international collaborative studies with set analytical approaches and methodologies are needed.

We are hopeful that the thoroughness of this review, with a detailed focus on key analytical criteria, will spur others to undertake comparable efforts to rigorously interrogate the available literature with respect to other diseases that lack validated (early) biomarkers or therapeutic targets that could effectively and fully halt disease progression (i.e., a cure). It will initially only be through such critical evaluation of the already available data that it will become apparent which approaches, analytical paradigms, and models are working and which have perhaps seen their useful period. This will then clarify the need for more critical approaches focused on substantially improved and better rationalized experimental designs, and with a key emphasis on data quality and proteoform identification, rather than sample throughput. It is the depth and specificity of the data that are important rather than the rate and sheer volume of its generation.

## Figures and Tables

**Figure 1 ijms-22-07377-f001:**
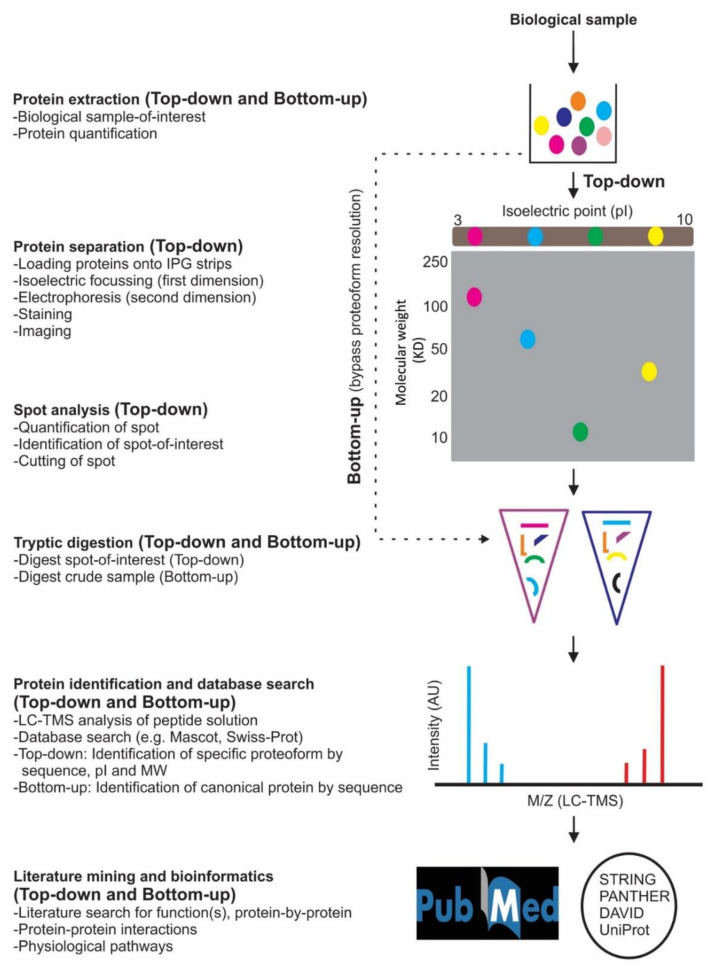
Schematic of top-down and bottom-up proteomic approaches. In top-down, proteoforms are extracted (sometimes then separated into total soluble and membrane fractions [104]) and resolved based on charge (i.e., pI) using immobilized pH gradient (IPG) strips, and then by size (i.e., ~MW) using SDS-PAGE gels. Spots of interest are excised from the gel, digested using proteases, and analysed by liquid chromatography-tandem mass spectrometry (LC-TMS). In contrast, bottom-up or shotgun approaches use crude digestion of the total proteoform extract, without prior resolution of intact proteoforms, and this peptide digest is then analysed by LC-TMS. In both approaches, database searches (e.g., Mascot, Swiss-Prot) then identify amino acid sequence matches; literature mining (PubMed) and bioinformatic analyses (e.g., STRING, DAVID, UniProt) are then used to assess potential functions, physiological pathways, and interactions. Image processing software CorelDraw-version 2018 (www.coreldraw.com, Ottawa, ON, Canada) was used to construct Figure.

**Table 1 ijms-22-07377-t001:** Use of different biological samples in proteomic analyses of MS, EAE, and CPZ.

Sample	MS	EAE	CPZ
CSF	[59,60,61,63,64,65,66,67,68,69,70,71,72,73,75,77,78,82,83,84]	[99]	-
Blood	[57,62,74,79,84,85]	[98]	[103]
Tear	[80]	-	-
Urine	[81]	-	-
Cerebrum	[58,76]	[86,90,92,95]	[101,102,103,104,105]
Cerebellum	-	[92]	-
Brain stem	-	[92,100]	-
Spinal cord	-	[87,88,89,92,93,94,95,96,97]	-
Spleen	-	-	[103]
Stool	-	[91]	-

Key: -, not found/no research. A full list of samples is provided in the Appendix A.

**Table 2 ijms-22-07377-t002:** Canonical proteins identified in different sample types from MS patients.

Canonical Proteins	Blood	Tear	Urine	CSF	Brain
14-3-3 protein	-	-	[81]	[67]	-
Actin	[62]	-	-	[63,64,75]	-
Albumin	[84]	[80]	-	[61,63,64,68,70,75]	-
Alpha-1-antichymotrypsin	-	[80]	-	[60,64,66,67,69,78]	-
Alpha-enolase	[62]	-	-	-	[76]
Annexin	[57,62]	[80]	-	[77]	-
Apolipoprotein	[79]	[80]	-	[59,60,63,64,69,70,71,72,75]	-
Brevican core protein	-	-	-	[60]	[76]
Clusterin	[79]	-	-	[64,69,70,75]	-
Complement	[79,84]	[80]	-	[63,64,69,70,72,75,84]	-
Contactin 1	-	-	-	[69,70,78]	[76]
Corticosteroid-binding globulin	[85]	-	-	[60]	-
Creatine kinase	[62]	-	-	[77]	[58]
Cystatin	[57]	[80]	-	[63,64,69,70,75,77]	-
Fatty acid-binding protein	[57]	[80]	-	-	-
Fibrinogen	[84]	-	-	[63,73,75,84]	-
Gelsolin	[79]	-	-	[63,64,70,72]	-
Glutathione S-transferase	[57]	[80]	-	-	-
Hemoglobin	-	-	[81]	[75]	[58]
Heat shock protein	[62]	[80]	-	-	-
Immunoglobulin	[57]	[80]	[81]	[60,63,64,70,75,83]	-
Lipocalin	-	[80]	-	[63]	-
Neutral alpha-glucosidase	[62]	[80]	-	-	-
Phosphatidylethanolamine binding protein	-	-	[81]	[63]	-
Protein S100	[57,85]	[80]	-	-	-
Receptor-type tyrosine-protein phosphatase	[57]	-	[81]	-	-
Secretogranin	[84]	-	-	[60,73,84]	-
Vitamin D-binding protein	[79]	-	-	[60,63,64,70,75,82]	-

Key: -, not found/no research. A full list of proteins is provided in the Appendix A. Proteins were selected from Appendix A and listed here when the same protein was found in the same (e.g., CSF) or different samples (e.g., CSF, blood). During selection, proteins were accepted regardless of the change in abundance (i.e., increase or decrease), sex, MS phenotype, or age.

**Table 3 ijms-22-07377-t003:** Canonical proteins identified in MS, EAE, and CPZ.

Canonical Proteins	Gene ID	Molecular Function	Experimental Group and Sample Analysed					
			MS		EAE		CPZ	
Methodology			TD	BU	TD	BU	TD	BU
2′-5′-oligoadenylate synthase	Oasl2	Metabolic	-	↓[57]; blood	-	↑[92]; spinal cord	-	↑[101]; cerebrum
5′(3′)-deoxyribonucleotidase	Nt5m	Metabolic	-	↑[57]; blood	↑[95]; cerebrum	↓[92]; spinal cord	-	-
Aconitate hydratase	Aco2	Metabolic	-	-	↑[100]; brain stem	↓[94]; spinal cord	↑[104]; cerebrum	-
Acyl carrier protein	Ndufab1	Metabolic	-	↑[57]; blood	-	↓[92]; spinal cord	-	-
Acyl-CoA synthetase	Acsm1	Metabolic	-	↑[57]; blood	-	↓[92]; spinal cord	-	↑[102,105]; cerebrum
Adenine phosphoribosyl transferase	Aprt	Metabolic	-	↑[57]; blood	-	↑[92]; spinal cord	-	-
Aldehyde dehydrogenase	Aldh2	Metabolic	↑[80]; tear	-	-	↑↓[92,94]; spinal cord	-	-
Aldose reductase	Akr1b1	Metabolic	-	-	-	↑[92]; spinal cord	↑↓[103]; cerebrum, spleen	-
Apolipoprotein	Apo	Metabolic	↑↓[59,63,64,71,72,75,79,80]; CSF, blood, tear	↑↓[60,69,70]; CSF	↑[88,89,95]; brain, spinal cord	↑↓[92,94,97,99]; cerebrum, spinal cord	-	↑[105]; cerebrum
Arginase-1	Arg1	Metabolic	-	-	-	↑[87]; spinal cord	↑[103]; spleen	-
Aspartate aminotransferase	Got1	Metabolic	-	-	↓[100]; brain stem	↓[92,94]; spinal cord	↓[104]; cerebrum	-
ATP synthase subunit	Atp5	Metabolic	-	-	↑[100]; brain stem	↑↓[92,94]; spinal cord	↑[104]; cerebrum	-
ATP-citrate synthase	Acly	Metabolic	-	-	-	↓[92,94]; spinal cord	↑[103]; cerebrum	-
cAMP-dependent protein kinase	Prkar	Metabolic	-	↑[57]; blood	-	↓[92]; spinal cord	-	-
Carbamoyl-phosphate synthase	Cps1	Metabolic	-	-	-	↓[92]; cerebrum	↑[103]; spleen	-
Carbonic anhydrase-2	Ca2	Metabolic	-	-	-	↑↓[92]; cerebrum, cerebellum	-	↓[105]; cerebrum
Ceruloplasmin	CP	Metabolic	↑↓[64,75]; CSF	↑↓[60,70]; CSF	-	↑[92,94,97]; spinal cord	-	-
Alpha-enolase	Enoa	Metabolic	↑[62]; blood	↑[76]; cerebrum	↑[88,100]; brain stem, spinal cord	↓[94]; spinal cord	-	-
Corticosteroid-binding globulin	Serpina6	Metabolic	-	↑[60,85]; CSF, blood	-	↑[99]; spinal cord	-	-
Creatine kinase	Ckb	Metabolic	↑[62]; blood	↑[58,77]; cerebrum, CSF	↑↓[88,89,95]; spinal cord	↓[94]; spinal cord	↑[104]; cerebrum	↑[102]; cerebrum
Dihydrolipoyl lysine-residue succinyl transferase	Dlst	Metabolic	-	-	↓[100]; brain stem	↓[92]; spinal cord	↓[103]; cerebrum	-
Cytochrome C oxidase	Cox	Metabolic	-	↓[58]; cerebrum	↓[89,90]; cerebrum, spinal cord	↑↓[92,94]; spinal cord	-	-
Dual specificity phosphatase	Dusp	Metabolic	-	↓[57]; blood	-	↓[92]; spinal cord	-	-
Dynactin	Dctn	Metabolic	↑[62]; blood	-	-	↑↓[92]; spinal cord	-	-
Glucosamine-6-phosphate isomerase	Gnpda	Metabolic	-	↑[57]; blood	-	↑[92]; spinal cord	-	-
Glutamate dehydrogenase	Glud	Metabolic	-	-	↑↓[96,100]; brain stem, spinal cord	↓[92,94]; spinal cord	↑[104]; cerebrum	-
Glutathione peroxidase	Gpx3	Metabolic	↑[63]; CSF	-	-	↑[92]; spinal cord	-	-
Glutathione S-transferase	GSTs	Metabolic	↑[80]; tear	↓[57]; blood	-	↑↓[92,94]; spinal cord	-	-
Glyceraldehyde-3-phosphate dehydrogenase	Gapdhs	Metabolic	-	-	↑[88]; spinal cord	↓[92]; spinal cord	-	↓[105]; cerebrum
Glycogen phosphorylase	Pygm	Metabolic	-	-	-	↓[92,94]; spinal cord	-	↑[105]; cerebrum
Hexokinase	Hk	Metabolic	-	-	-	↑↓[92,94]; spinal cord	↓[104]; cerebrum	-
Ubiquitin carboxyl terminal hydrolase	Ubp	Metabolic	↓[75]; CSF	-	-	↓[92]; cerebrum	-	-
L-lactate dehydrogenase B chain	Ldhb	Metabolic	↑[63]; CSF	-	↑[100]; brain stem	↓[94]; spinal cord	-	-
Malate dehydrogenase	Mdh2	Metabolic	-	-	↑[88,89,100]; brain stem, spinal cord	↓[92,94]; spinal cord	↑[104]; cerebrum	-
NAD-dependent protein deacetylase sirtuin-2	Sirt2	Metabolic	-	-	-	↑↓[92,94]; spinal cord	↓[104]; cerebrum	-
NADH dehydrogenase [ubiquinone] 1 α	Ndufa	Metabolic	-	-	-	↓[92]; spinal cord	↑[104]; cerebrum	↑[105]; cerebrum
NADH dehydrogenase iron-sulfur protein	Ndufs	Metabolic	-	↑[57]; blood	-	↑↓[92,94]; spinal cord	↑[104]; cerebrum	-
Myeloblastin	Prtn3	Metabolic	-	↓[57]; blood	-	↓[87]; blood	-	-
Peptidyl-prolyl cis–trans isomerase	Ppi	Metabolic	-	↓[57]; blood	-	↑↓[92,94,97]; spinal cord	-	-
Peroxiredoxin	Prdx	Metabolic	-	↑[76]; cerebrum	↑↓[89,95,100]; cerebrum, brain stem, spinal cord	↑↓[92,94]; spinal cord	-	↑[105]; cerebrum
Phosphatidylinositol 3-kinase	Pik3r	Metabolic	-	↑[57]; blood	-	↓[92]; spinal cord	-	-
Phosphoglycerate kinase 1	Pgk1	Metabolic	↑[62]; blood	-	↑[88]; spinal cord	↓[94]; spinal cord	-	-
Prostaglandin-H2 D-isomerase	Ptgds	Metabolic	↑[66]; CSF	-	-	↑[92]; spinal cord	-	-
Protein phosphatase 1 regulatory subunit	PP1R	Metabolic	-	-	↑↓[89,90]; cerebrum, spinal cord	↑↓[92]; spinal cord	-	↑[102]; cerebrum
Pyruvate kinase isozymes M1/M2	Kpym	Metabolic	↑[62]; blood	-	-	↑[94]; spinal cord	-	-
Receptor-type tyrosine-protein phosphatase	Ptprj	Metabolic	-	↓[57,81]; blood, urine	-	↑↓[92]; spinal cord	-	-
Paraoxonase/arylesterase 1	Pon1	Metabolic	-	↑[60]; CSF	-	↑[92]; spinal cord	-	-
Superoxide dismutase	Sod	Metabolic	↑↓[63,64,75,78]; CSF	-	↓[95]; spinal cord	↑↓[92,94]; cerebrum, spinal cord	-	↓[105]; cerebrum
Tyrosine-protein phosphatase non-receptor type	Ptpn	Metabolic	-	-	↑[100]; brain stem	↑[87,92]; spinal cord	↑[103]; spleen	-
Transketolase	Tkt	Metabolic	-	↑[57]; blood	↑[88]; spinal cord	-	-	↓[105]; cerebrum
Actin	Actg	Structural	↑↓[62,63,64,75]; blood, CSF	-	↑↓[88,95,100]; cerebrum, brain stem, spinal cord	↑↓[92,94]; spinal cord	↓[104]; cerebrum	-
Cofilin 1	Cof1	Structural	↓[62]; blood	-	-	↓[94]; spinal cord	-	-
Collagen alpha-1(I) chain	Co1a1	Structural	-	↓[81]; urine	-	↑↓[92,94]; spinal cord	↑[103]; spleen	↑[102]; cerebrum
Brevican core protein	Bcan	Structural	-	↓[60,76]; CSF, cerebrum	-	↓[92]; spinal cord	-	-
Cadherin	Cad	Structural	-	↑[67]; CSF	-	↓[92]; spinal cord	-	-
Cell adhesion molecule	Cadm1	Structural	↑[64]; CSF	-	-	↓[92]; spinal cord	-	-
Alpha-adducin	Add1	Structural	-	↑[57]; blood	-	↓[92]; spinal cord	-	-
ADP-ribosylation factor 4	Arf4	Structural	-	↓[57]; blood	-	-	-	↑[105]; cerebrum
Coronin-1A	Coro1a	Structural	-	↑[57]; blood	↑[88]; spinal cord	-	-	-
Cytokeratin	Krt	Structural	↑[68]; CSF	-	↓[90]; cerebrum	-	-	-
Desmoplakin	Dsp	Structural	↑[66]; CSF	-	↑[100]; brain stem	-	-	-
DnaJ homolog subfamily C member 1	Dnajc	Structural	-	-	-	↑↓[92]; spinal cord	↑[103]; spleen	-
Fibulin	Fbln	Structural	↑↓[64,78]; CSF	-	-	↑[92]; spinal cord	-	-
Filamin A	Flna	Structural	-	↑[57]; blood	-	↑[92,97]; spinal cord	-	-
Integrin	Itg	Structural	-	↓[57]; blood	-	↑↓[87,92]; blood, cerebrum, spinal cord	-	↑[101]; cerebrum
Intercellular adhesion molecule 1	Icam1	Structural	-	-	-	↑[92]; spinal cord	-	↑[101]; cerebrum
Lysosome-associated membrane glycoprotein	Lamp	Structural	-	↑[81]; urine	-	↑[92]; spinal cord	-	-
Lamin	Lmn	Structural	-	↑[57]; blood	↑[88]; spinal cord	↑[92]; spinal cord	-	-
Myosin	Myh	Structural	↑[63]; CSF	-	↓[90]; cerebrum	↑[92,94,97]; spinal cord	↑↓[103]; spleen	↑[105]; cerebrum
Prelamin-A/C	Lmna	Structural	-	-	-	↑[92]; spinal cord	↑[103]; spleen	-
Ribosome-binding protein 1	Rrbp1	Structural	-	-	-	↑[92]; spinal cord	↓[103]; spleen	-
Septin	Sept	Structural	↓[62]; blood	-	↑↓[89,90,95,100]; cerebrum, brain stem, spinal cord	↑↓[92,94]; spinal cord	↓[104]; cerebrum	-
Transmembrane protein	Tmem	Structural	-	↑[67]; CSF	-	↑↓[92]; cerebrum, spinal cord	-	-
Tubulin	Tub	Structural	↑[62]; blood	-	↑↓[88,95,100]; cerebrum, brain stem, spinal cord	↑↓[92,94]; spinal cord	↓[103]; spleen	-
Thymosin beta-4	Tmsb4x	Structural	-	↓[73]; CSF	-	↓[92]; cerebrum	-	-
Vinculin	Vinc	Structural	↓[62]; blood	↓[57]; blood	-	↑[92]; spinal cord	-	-
Zinc finger protein	Zn	Structural	↓[75]; CSF	↓[82]; CSF	-	↑↓[92]; cerebrum	-	-
Annexin	Anxa	Immune response	↑[62,80]; blood, tear	↑↓[57,77]; blood, CSF	↑[88,89,100]; brain stem, spinal cord	↑↓[87,92,94,97]; blood, cerebrum, spinal cord	-	↑[105]; cerebrum
Complement (e.g., C3, C4)	C3	Immune response	↑↓[63,64,72,75,79,80]; CSF, blood, tear	↑↓[69,70,84]; CSF, blood, cerebrum, cerebellum, brain stem, spinal cord	-	↑↓[87,92,97,99]; spinal cord	↓[103]; blood	-
Dedicator of cytokinesis	Doc	Immune response	-	↑[57]; blood	-	↑↓[92]; cerebrum, spinal cord	-	-
Gasdermin-D	Gsdmd	Immune response	-	↓[57]; blood	-	↓[92]; spinal cord	-	-
Glial fibrillary acidic protein	Gfap	Immune response	-	↑[76]; cerebrum	↑[88,89,96,100]; brain stem, spinal cord	↑[92,94]; spinal cord	↑[104]; cerebrum	↑[105]; cerebrum
Immunoglobulin	Ig	Immune response	↑↓[59,61,63,64,71,75,80]; CSF, tear	↑↓[57,60,70,81,83]; blood, CSF, urine	-	↑↓[92,97]; cerebrum, brain stem, spinal cord	-	↑[102,105]; cerebrum
Interferon-induced 35 kDa protein	IN35	Immune response	↑[62]; blood	-	-	↑[92]; brain stem	-	-
Macrophage migration inhibitory factor	Mif	Immune response	-	↑[57]; blood	-	-	-	↓[105]; cerebrum
Monocyte differentiation antigen CD14	CD14	Immune response	-	↓[70]; CSF	-	↑[92]; spinal cord	-	-
Neuronal cell adhesion molecule	Nrcam	Immune response	↓[78]; CSF	↑[67]; CSF	-	↓[92,94]; spinal cord	-	-
Nuclear factor NF-kappa-B	Nfkb	Immune response	-	↑[57]; blood	-	↑[92]; spinal cord	-	-
Osteopontin	Spp1	Immune response	-	↑[67]; CSF	-	↑[92]; spinal cord	-	↑[101]; cerebrum
Protein S100	S100	Immune response	↑[80]; tear	↑↓[57,85]; blood	-	-	-	↑[102,105]; cerebrum
Ras-related C3 botulinum toxin substrate 3	Rac1	Immune response	-	-	-	↑↓[92]; spinal cord	-	↑[105]; cerebrum
Ras-related protein Rab	Rab	Immune response	-	-	-	↑↓[92,94]; spinal cord	-	↑[105]; cerebrum
T-complex protein	Tcp	Immune response	-	-	↑[89]; spinal cord	-	↑[103]; spleen	-
Tumor necrosis factor-α	Tnf	Immune response	-	↑[57]; blood	-	↑↓[92]; spinal cord	-	-
Toll-like receptor	Tlr	Immune response	-	-	-	↑[92]; spinal cord	-	↑[101]; cerebrum
Vimentin	Vim	Immune response	-	↑[77]; CSF	↑[89]; spinal cord	↑[92,94,97]; cerebrum, spinal cord	-	↑[105]; cerebrum
Albumin	Alb	Blood-related	↑↓[61,63,64,68,75,80]; CSF, tear	↓[70,84]; CSF, blood	↑[88,96,100]; cerebrum, brain stem, spinal cord	↑↓[87,92,97]; spinal cord	-	-
Alpha-2-HS-glycoprotein	Ahsg	Blood-related	-	↓[70]; CSF	-	↑[87]; spinal cord	-	-
Antithrombin	Serpinc1	Blood-related	↑[64,75]; CSF	↓[70]; CSF	↑[91]; stool	-	-	-
Beta-2-microglobulin	B2m	Blood-related	↑[64,75,176]; CSF	↑[82]; CSF	-	↑[92]; spinal cord	-	-
Chitinase-3 like protein 1	Chi3l1	Blood-related	↑[64]; CSF	↑[60]; CSF	-	↑[87,92]; spinal cord	-	-
Haptoglobin	Hp	Blood-related	↑↓[63,64,72,75]; CSF	↑[60,82]; CSF	↑[90]; cerebrum	-	-	-
Fibrinogen	Fgl1	Blood-related	↑↓[63,75]; CSF	↑[73,84]; CSF, blood	↑↓[89,95]; cerebrum, spinal cord	↑↓[92,94,99]; cerebrum, spinal cord	-	-
Hemoglobin	Hb	Blood-related	-	↑[58,81]; cerebrum, urine	-	↓[92,94]; cerebrum, spinal cord	-	-
Hemopexin	Hpx	Blood-related	↑[79]; blood	-	↑[88,89]; spinal cord	↑↓[92,94,97]; cerebrum, spinal cord	-	-
Macroglobulin-α2	A2m	Blood-related	↑[63]; CSF	-	-	↑[97]; spinal cord	-	-
Plasminogen	Plmn	Blood-related	↑↓[64,65,75]; CSF	↑↓[60,77]; CSF	-	↑[92]; spinal cord	-	-
Prothrombin	F2	Blood-related	↑[64]; CSF	-	-	↑[99]; spinal cord	-	-
Serotransferrin	Tf	Blood-related	↑[75]; CSF	↑↓[69,70,77]; CSF	↑[96,100]; brain stem, spinal cord	↑↓[87,92,97]; blood, spinal cord	-	-
Thrombospondin 1	Thbs1	Blood-related	-	↑[85]; blood	-	↑[87]; blood	-	-
Transthyretin	Ttr	Blood-related	↑↓[63,64,67,71,75]; CSF	-	-	↓[92]; cerebrum	-	-
Vitamin D binding protein	Gc	Blood-related	↑↓[63,64,75,79]; CSF, blood	↑↓[60,70,82]; CSF	-	↑[87]; spinal cord	-	-
Galectin-related protein	Lgalsl	Signalling	-	↓[57]; blood	-	↓[92]; spinal cord	-	-
Guanine nucleotide binding protein	Gnao	Signalling	-	↑↓[57]; blood	↑↓[86,100]; cerebral microvessel, brain stem	↑↓[92,94]; spinal cord	↑[104]; cerebrum	↓[105]; cerebrum
LIM and senescent cell antigen-like domains 1	Lims1	Signalling	-	↓[57]; blood	-	↑[92]; spinal cord	-	-
Myristoylated alanine-rich C-kinase substrate	Marcks	Signalling	-	-	-	↑[92,94]; spinal cord	-	↓[105]; cerebrum
40S ribosomal protein S3	Rps	Signalling	-	-	-	↑[92,94,97]; spinal cord	-	↑[105]; cerebrum
AP2-associated protein kinase 1	AAK1	Signalling	-	↑[57]; blood	-	↓[92]; spinal cord	-	-
Calcium/calmodulin-dependent protein kinase	Camk	Signalling	-	↑[57]; blood	-	↓[92,94]; spinal cord	↓[104]; cerebrum	↑[101,105]; cerebrum
Protein kinase C	Prkc	Signalling	-	-	-	↑↓[92]; cerebellum, spinal cord	-	↑[105]; cerebrum
Regulator of G-protein signalling	Rgs	Signalling	-	↑[57]; blood	-	↑[92]; cerebellum, spinal cord	-	-
Thioredoxin	Thio	Signalling	↑[62]; blood	-	-	↑↓[92,94]; spinal cord	-	-
Voltage-dependent anion-selective channel protein	Vdac2	Signalling	-	-	-	↑↓[92,94]; cerebrum, spinal cord	↓[104]; cerebrum	-
14-3-3 protein epsilon	Ywhae	Myelin component	-	↑↓[67,81]; CSF, urine	↑↓[89,90]; cerebrum, spinal cord	↓[94]; spinal cord	↓[104]; cerebrum	-
Amyloid beta	App	Myelin component	↑[64,79]; CSF, blood	↓[70]; CSF	↑[91,95]; spinal cord, stool	↑↓[92]; cerebrum, spinal cord	-	-
Contactin 1	Cntn1	Myelin component	↑[78]; CSF	↓[69,70,76]; CSF, cerebrum	-	↓[92]; spinal cord	-	↓[105]; cerebrum
Myelin basic protein	Mbp	Myelin component	-	↑[58]; cerebrum	-	↑↓[87,92,94]; cerebrum, cerebellum, brain stem, spinal cord	-	↓[105]; cerebrum
Myelin proteolipid protein	Plp	Myelin component	-	-	-	↑[94]; spinal cord	-	↓[105]; cerebrum
Myelin-associated glycoprotein	Mag	Myelin component	-	↓[76]; cerebrum	-	↓[92,94]; spinal cord	-	↓[105]; cerebrum
Myelin-associated oligodendrocytic basic protein	Mobp	Myelin component	-	-	-	↓[94]; spinal cord	-	↓[105]; cerebrum
Neurofilament	Nef	Myelin component	↑[75]; CSF	↑[67]; CSF	↑[88]; spinal cord	↑↓[92,93,94,98]; blood, spinal cord	↓[104]; cerebrum	-
Alpha-1-antitrypsin	Serpina1a	Protease inhibitor	↑↓[64,75]; CSF	-	↑[91]; stool	-	-	-
Angiotensinogen	Agt	Protease inhibitor	↑↓[64,78]; CSF	-	-	↑[92]; spinal cord	-	-
Cystatin (e.g., A)	Cyta	Protease inhibitor	↑↓[63,64,75,80]; CSF, tear	↑↓[57,69,70,77]; blood, CSF	↑[95]; spinal cord	↑↓[92,97]; spinal cord	-	-
Phosphatidylethanolamine binding protein	Pebp	Protease inhibitor	↑[63]; CSF	↓[81]; urine	↑↓[88,90]; cerebrum, spinal cord	↓[94]; spinal cord	-	-
Serine proteinase inhibitor	Serpina	Protease inhibitor	↓[71]; CSF	-	↑[91]; stool	-	-	-
Leukocyte elastase inhibitor A	Serpinb1a	Protease inhibitor	-	-	↑[100]; brain stem	↓[92]; spinal cord	↑↓[103,104]; spleen, cerebrum	-
Calnexin	Calx	Molecular chaperone	-	↓[57]; blood	-	↑[94]; spinal cord	-	-
Calreticulin	Calr	Molecular chaperone	-	-	↓[89,90]; cerebrum, spinal cord	↑[92,94]; spinal cord	↑[104]; cerebrum	-
Clusterin	Clu	Molecular chaperone	↑↓[64,75,79]; CSF, tear	↑↓[69,70]; CSF	-	↑[92]; spinal cord	-	-
Heat shock protein	Hsp	Molecular chaperone	↑[62,80]; blood, tear	-	↑↓[88,89,95,100]; brain stem, spinal cord	↑↓[87,92,94]; spinal cord	-	↓[105]; cerebrum
Protein disulfide-isomerase	Pdia	Molecular chaperone	↑[62]; blood	↓[57]; blood	↑↓[88,96,100]; brain stem, spinal cord	↑[92]; spinal cord	↓[103]; spleen	-
Ubiquitin-like protein ISG15	Isg15	Molecular chaperone	-	-	-	↓[92]; spinal cord	-	↑[101]; cerebrum
Cathepsin	Cts	Protease	-	-	-	↑[92,97]; cerebrum, spinal cord	-	↑[105]; cerebrum
Chromogranin-A	Chga	Protease	-	↓[69]; CSF	-	↓[92]; cerebrum	-	-
Vitronectin	Vtn	Protease	↑[64]; CSF	↑[60]; CSF	-	↑[92]; spinal cord	-	-
Kallikrein 6	Klk6	Protease	↑↓[63,64,65,75]; CSF	↓[69,70]; CSF	-	↑[92]; spinal cord	-	-
Charged multivesicular body protein	Chmp4b	Exocytosis	-	↑↓[57]; blood	-	↓[92]; cerebrum, spinal cord	↓[104]; cerebrum	-
Clathrin light chain A	Cltc	Exocytosis	-	↓[57]; blood	-	↑↓[92,94]; spinal cord	-	-
Syntaxin-binding protein	Stxbp	Exocytosis	-	↑[57]; blood	↑[96,100]; brain stem, spinal cord	↓[92,94]; spinal cord	↓[104]; cerebrum	-
Vesicle-fusing ATPase	Nsf	Exocytosis	-	-	-	↓[92,94]; spinal cord	↓[104]; cerebrum	-
Amphiphysin	Amph	Endocytosis	-	-	-	↓[92]; spinal cord	-	↑[105]; cerebrum
Dynamin 1	Dnm1	Endocytosis	-	-	↓[100]; spinal cord	↓[92,94]; spinal cord	↓[103,104]; cerebrum	-
Elongation factor 2	Eef2k	Translation	-	-	↑[100]; spinal cord	-	-	↑[105]; cerebrum
Heterogeneous nuclear ribonucleoprotein	Hnrp	Translation	↓[62]; blood	-	↑[86]	↑↓[92,94]; spinal cord	-	-
Fatty acid-binding protein	Fabp5	Transportation	↑[80]; tear	↓[57]; blood	↓[89]; spinal cord	↑↓[92,94]; cerebrum, spinal cord	-	-
Sideroflexin	Sfxn	Transportation	-	-	-	↓[92]; spinal cord	-	↑[105]; cerebrum
Signal transducer and activator of transcription	Stat	Transcription	-	-	-	↑[87,92]; spinal cord	-	↑[101]; cerebrum
Paired amphipathic helix protein Sin3a	Sin3a	Transcription	-	↑[57]; blood	-	↑[92]; spinal cord	-	-
Rab GDP dissociation inhibitor	Gdi	Neurotransmission	-	-	-	↓[94]; spinal cord	↓[104]; cerebrum	-
Synaptosomal-associated protein	Snap	Neurotransmission	-	-	↓[89,90]; cerebrum, spinal cord	↓[92,94]; spinal cord	-	↓[101]; cerebrum
Copine	Cpne	Binding	-	↑[57]; blood	-	↑↓[92]; spinal cord	-	-
Caspase	Casp	Apoptosis	-	↑[77]; CSF	-	↑[92]; spinal cord	-	-

Key: ↑, increase; ↓, decrease; MS, Multiple Sclerosis; EAE, Experimental Autoimmune Encephalomyelitis; CPZ, Cuprizone; TD, Top-down; BU, Bottom-up and -, not found/no research. Studies that mentioned only the presence of a protein without describing the magnitude of change (e.g., fold increase or decrease) compared to Controls are indicated with a ↑ sign. On the other hand, if a protein was described as absent, a ↓ sign is used, to maintain the consistency with other studies. A full list of proteins is provided in the Appendix A.

**Table 4 ijms-22-07377-t004:** Changes in canonical proteins in different subtypes of MS.

Canonical proteins	Molecular Function	PPMS	SPMS	RRMS	UMS	CIS	PMS
Albumin	Blood-related	↑↓[80,84]	↑↓[63,80,84]	↑↓[61,63,64,68,70,75,80]	-	-	-
Alpha 2-HS glycoprotein	Blood-related	-	↑[65]	↑↓[65,70,71]	-	↓[71]	-
Alpha-1-acid glycoprotein	Blood-related	-	-	↓[69]	-	↓[69]	↑[79]
Alpha-1B glycoprotein	Blood-related	-	-	↑↓[70,75]	-	-	↑[79]
Alpha-2-macroglobulin	Blood-related	-	-	↓[69,70]	-	↓[69]	-
Beta-2-microglobulin	Blood-related	↑[65,82]	↑[65]	↑[64,75,82]	-	-	-
Chitinase-3 like protein 1	Blood-related	-	-	↑[64]	-	↑[60]	-
Corticosteroid-binding globulin	Blood-related	-	-	-	↑[85]	↑[60]	-
Fibrinogen	Blood-related	↑[73,84]	↑[63,84]	↑↓[63,73,75,84]	-	↑[73]	-
Haptoglobin	Blood-related	↓[80]	↑↓[63,80]	↑↓[63,64,72,75,80,83]	-	↑[60]	-
Pigment epithelium derived factor	Blood-related	-	↑[63]	↑↓[63,64,75]	-	-	-
Plasminogen	Blood-related	↓[65]	↓[65]	↑↓[64,75,77]	-	↑[60]	-
Platelet basic protein	Blood-related	-	-	↓[57]	↑[85]	-	-
Serotransferrin	Blood-related	-	-	↑↓[69,70,75,77]	-	↑[69]	-
Transferrin	Blood-related	↓[65]	↑↓[63,65]	↑↓[63,64,71]	-	↓[71]	-
Transthyretin	Blood-related	-	↑[63]	↑↓[63,64,71,75]	-	↓[71]	-
Vitamin D binding protein	Blood-related	↓[82]	↑[63]	↑↓[63,64,70,75,82]	-	↑[60]	↑[79]
Alpha-enolase	Metabolic	-	-	↑[62]	↑[76]	-	-
Apolipoprotein	Metabolic	↑[80]	↑[63,80]	↑↓[59,63,64,69,71,72,75,80]	-	↑↓[60,69,71]	↑[79]
Beta-Ala-His dipeptidase	Metabolic	-	-	↓[70]	-	↓[60]	-
Ceruloplasmin	Metabolic	-	-	↑[64,70,75]	-	↓[60]	-
Creatine kinase	Metabolic	-	-	↑[62,77]	↑[58]	-	-
Glutathione S-transferase	Metabolic	↑[80]	↑[80]	↑↓[57,80]	-	-	-
Prostaglandin D2-synthase	Metabolic	-	↑[63]	↑[64,66]	-	-	-
Receptor-type tyrosine-protein phosphatase	Metabolic	-	-	↓[81]	↓[57]	-	-
Superoxide dismutase	Metabolic	-	↑↓[63,78]	↑↓[63,64,75,78]	-	-	-
Annexin	Immune response	↑[80]	↑[80]	↑↓[57,62,77,80]	-	-	-
Complement	Immune response	↑[80,84]	↑[63,80,84]	↑↓[63,64,69,70,72,75,80,84]	-	↓[69]	-
Immunoglobulin	Immune response	↑[80]	↑[63,80]	↑↓[57,59,61,63,64,70,71,75,80,83]	↓[81]	↑↓[60,71]	-
Lipocalin	Immune response	↑[63,80]	↑[80]	↑[63,80]	-	-	-
Neuronal cell adhesion molecule	Immune response	-	↑↓[67,78]	↓[78]	-	-	-
Protein S100	Immune response	↑[80]	↑[80]	↑↓[57,80]	↑[85]	-	-
Actin	Structural	-	↑[63]	↑↓[62,63,64,75]	-	-	-
Brevican core protein	Structural	-	-	-	↓[76]	↓[60]	-
Fibulin 1	Structural	-	↓[78]	↑↓[64,78]	-	-	-
Gelsolin	Structural	-	↑[63]	↑↓[63,64,70,72]	-	-	↑[79]
Zinc finger protein	Structural	↓[82]	-	↓[75,82]	-	-	-
Kallikrein 6	Protease	↓[65]	↑↓[63,65]	↑↓[63,64,69,70,75]	-	↓[69]	-
Vitronectin	Protease	-	-	↑[64]	-	↑[60]	-
Antichymotrypsin	Protease inhibitor	↑[80]	↑[67,78,80]	↑↓[64,66,69,78,80]	-	↓[60,69]	-
Angiotensinogen	Protease inhibitor	-	↓[78]	↑↓[64,78]	-	-	-
Cystatin	Protease inhibitor	↑[80]	↑[63,80]	↑↓[57,63,64,69,70,75,77,80]	-	↓[69]	-
Phosphatidylethanolamine binding protein	Protease inhibitor	-	↑[63]	↑[63]	↓[81]	-	-
14-3-3 protein	Myelin component	-	↑[67]	-	↓[81]	-	-
Contactin 1	Myelin component	-	↑[78]	↑↓[69,70,78]	↓[76]	↓[69]	-
Clusterin	Molecular chaperone	-	-	↑↓[64,69,70,75]	-	↑[69]	↑[79]
Heat shock protein	Molecular chaperone	↑[80]	↑[80]	↑[62,80]	-	-	-
Fatty acid-binding protein	Transportation	↑[80]	↑[80]	↑↓[57,80]	-	-	-
Secretogranin	Exocytosis	↓[73,84]	↓[84]	↓[73,84]	-	↓[60,73]	-

Key: -, not found/no research; ↑, increase; ↓, decrease; PPMS, primary progressive MS; SPMS, secondary progressive MS; RRMS, relapsing-remitting MS; UMS, uncategorized MS; CIS, clinically isolated symptoms; PMS, pediatric MS. Studies that mentioned only the presence of proteins without describing the magnitude of change (e.g., fold increase or decrease) compared to Controls are considered as increase and a ↑ sign is used to indicate them. On the other hand, if is the protein was described as absent, a ↓ sign is used, to maintain the consistency with other studies. A full list of proteins is provided in the Appendix A.

## Data Availability

Appendix A associated with this manuscript are provided as separate attachments.

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
