# Peer review of "Proteomics of Multiple Sclerosis: Inherent Issues in Defining the Pathoetiology and Identifying (Early) Biomarkers"

_ijms, 2021, doi:10.3390/ijms22147377_

Round 1

Reviewer 1 Report

The authors summarized the available proteomic studies of MS patients and animal models using literature mining and bioinformatic analyses. The author drew a lot of conclusions. This review is more appropriate for a section of a doctoral thesis rather than an academic journal. The authors should not spend too much time on describing the common knowledge (e.g., proteomics methods), but ignoring the topic (MS) in this manuscript. In addition, the authors should summarize the conclusions concisely, and propose new insights and future directions of the early biomarkers of MS.

Author Response

Reviewer’s comments and response

We thank the reviewers for their efforts and constructive suggestions. We have revised accordingly. Newly added words/phrases are highlighted in yellow. We are grateful as the feedback has clearly improved the quality of our manuscript. We hope that this revised version will be acceptable for publication.

Reviewer 1:

Comment

The authors summarized the available proteomic studies of MS patients and animal models using literature mining and bioinformatic analyses. The author drew a lot of conclusions. This review is more appropriate for a section of a doctoral thesis rather than an academic journal. The authors should not spend too much time on describing the common knowledge (e.g., proteomics methods), but ignoring the topic (MS) in this manuscript. In addition, the authors should summarize the conclusions concisely, and propose new insights and future directions of the early biomarkers of MS.

Response:

We thank the reviewer for their comments. We mention at the beginning of the manuscript (line: 115) that there is no comprehensive review concerning application of proteomics to the field of Multiple Sclerosis (MS) research, and our aim is to provide a comprehensive systematic overview of proteomic studies in order to move the field forward by identifying research gaps. We thus feel the methods section (which amounts to only ~1 page in the current format, plus a figure) is critical since it is apparent in the literature that many MS researchers (or those focused more broadly on a range of diseases) are unfamiliar with the critical differences in approaches and thus think that all proteomic analyses provide equivalent data. This is simply not the case, and such misconceptions will not lead to the identification of critical proteoforms that can most effectively be used as biomarkers or for the targeted development of therapeutics. Nonetheless, overall, we have sought to condense the document as suggested by both reviewers. During submission of the manuscript, the total word count was 27088; now the total word count is 25058 which also includes the addition of sections from reviewer’s comments.   

In the Conclusion, we have further summarized our findings and made some future recommendations; these largely appear following the discussion of individual sections. In the manuscript, future needs/directions are highlighted in Green.

Lines: 104-106, 117-118, 125-126, 130, 253-254, 260-261, 265-266, 296-297, 395-396, 492-493, 656-657, 722-723, 725-726, 788-789, 885-891, 929-930, 947-951, 982-983.

Reviewer 2 Report

This manuscript is review of biomarkers for Multiple Sclerosis (MS) and the authors try to discuss the importance of unifying data criteria though the review. The approach is important and interesting. The manuscript is well constructed, but a bit long and redundant. It would be preferable to make it more compact for the readers.

I have a few suggestions.

  1. Some abbreviations (e.g. CPZ or MW) are not explained where they appear at the first time in the manuscript. It would be more helpful to explain them in the sentence as well as end.
  2. Figure 1 is confusing, and the difference between top-down and bottom-up proteomic approach is key point of the manuscript, so please modify Figure 1 to make it more intuitive for readers.
  3. Please explain the statements in 203 to 206 (Therefore, while the shotgun approach has gained substantial popularit ~) in more detail using specific examples.
  4. Please explain more about Proteoforms. Could proteins those proteoforms are different and translated from the same genome play different roles in a certain disease. In other words, is it possible that 3D structure or modification of a protein, rather than its genome, plays an important role in a disease? I would like you to explain with specific examples, even for diseases other than MS.
  5. I think the exact expression is good but this manuscript is a bit verbose in its expression. For example, the description of the definition of biomarker ( line 253 to 255). This may be necessary for the later discussion, but I think it is not essential. More straight forward expressions are helpful to readers. Please improve it.

Author Response

Reviewer’s comments and response

We thank the reviewers for their efforts and constructive suggestions. We have revised accordingly. Newly added words/phrases are highlighted in yellow. We are grateful as the feedback has clearly improved the quality of our manuscript. We hope that this revised version will be acceptable for publication.

Reviewer 2:

This manuscript is review of biomarkers for Multiple Sclerosis (MS) and the authors try to discuss the importance of unifying data criteria though the review. The approach is important and interesting. The manuscript is well constructed, but a bit long and redundant. It would be preferable to make it more compact for the readers.

Response:

We thank the reviewer for their comments. In the revised version of the manuscript, we provide point-by-point responses, as outlined below. In addition, we have reduced the amount of detail where relevant. During submission of the manuscript, the total word count was 27088; now the total word count is 25058 which also includes the addition of sections from reviewer’s comments.   

Comment

Some abbreviations (e.g. CPZ or MW) are not explained where they appear at the first time in the manuscript. It would be more helpful to explain them in the sentence as well as end.

Response:

Thank you for this comment. We have revised the whole manuscript in this regard, including the CPZ and MW instances noted. Lines: 73, 144

Comment

Figure 1 is confusing, and the difference between top-down and bottom-up proteomic approach is key point of the manuscript, so please modify Figure 1 to make it more intuitive for readers.

Response:

We thank the reviewer for this feedback and for emphasizing the importance of clearly differentiating between the two analytical approaches. We have now revised Figure 1 to better clarify these differences. Page: 5

Comment

Please explain the statements in 203 to 206 (Therefore, while the shotgun approach has gained substantial popularit ~) in more detail using specific examples.

Response:

We thank the reviewer for this comment. While seeking not to further add to the methods (see Reviewer 1), we have added a straightforward comment concerning the popularity of shotgun proteomic approach. Lines: 349-355

However, the routine bottom-up/shotgun approach requires less sample than top-down but only infers canonical protein identifications, providing no information regarding proteoforms. The shotgun approach thus involves bulk digestion of a total protein extract followed by mass spectrometric analysis (Figure 1). In contrast, the top-down proteomic approach uses two steps to resolve intact proteoforms prior to selective proteolytic digestion and mass spectrometric analysis (Figure 1). Therefore, while the bottom-up approach is often claimed to be ‘faster’, this is often by ignoring technical replicates and does not take into account the many multiple separate analyses that must occur in this approach for every potential PTM if critical information concerning proteoforms is even sought [1-3].

Comment

Please explain more about Proteoforms. Could proteins those proteoforms are different and translated from the same genome play different roles in a certain disease. In other words, is it possible that 3D structure or modification of a protein, rather than its genome, plays an important role in a disease? I would like you to explain with specific examples, even for diseases other than MS.

Response:

We thank you for this important comment and direction. While this is clearly the case, we had not emphasized this point specifically. We have now discussed the role of proteoforms in disease. Lines:  151-164

Many studies have revealed the association of proteoforms in the pathogenesis of MS. Kin et al. found an elevation of mono and dimethylated arginine but a reduction of phosphorylation in MS white matter samples [4]. Consistent with this, deimindaton, or citrullination, results in a conformational change when the amino acid arginine in myelin basic protein is converted to citrulline, a non-standard amino acid; this has been reported in MS lesions [5]. In contrast, these modifications were not found in non-MS human subjects [4,5]. Similar to the detection of this PTM in samples from human MS patients [4,5], citrullination has also been associated with autoimmune encephalomyelitis in the CPZ [6] and EAE animal models [7]. Notably, citrullination has also been linked to rheumatoid arthritis [8], and there is also evidence of an increased incidence of rheumatoid arthritis in MS patients [9]. Specific PTM have also been associated with other neurological diseases such as Alzheimer’s disease, and with traumatic brain injury [10,11]. In the latter case, following traumatic brain injury, tau acetylation resulted in neurodegeneration and neurobehavioral impairment in an animal model; inhibition of acetylation reversed the pathological outcome [11]. While far from exhaustive, these examples highlight that proteoforms are the functional, biologically active molecules, and emphasize the critical need to identify specific proteoforms in order to understand the native molecular mechanisms and thus also the pathophysiology of a disease.

Comment

I think the exact expression is good but this manuscript is a bit verbose in its expression. For example, the description of the definition of biomarker (line 253 to 255). This may be necessary for the later discussion, but I think it is not essential. More straight forward expressions are helpful to readers. Please improve it.

Response:

The exact expression has been removed and revised as- Line: 227

Biomarkers are indicators and predictors of particular (patho)physiological processes.

References

  1. Coorssen, J.R.; Yergey, A.L. Proteomics Is Analytical Chemistry: Fitness-for-Purpose in the Application of Top-Down and Bottom-Up Analyses. Proteomes 2015, 3, 440-453, doi:10.3390/proteomes3040440.
  2. Oliveira, B.M.; Coorssen, J.R.; Martins-de-Souza, D. 2DE: the phoenix of proteomics. J Proteomics 2014, 104, 140-150, doi:10.1016/j.jprot.2014.03.035.
  3. Zhan, X.; Li, B.; Zhan, X., et al. Innovating the Concept and Practice of Two-Dimensional Gel Electrophoresis in the Analysis of Proteomes at the Proteoform Level. Proteomes 2019, 7, doi:10.3390/proteomes7040036.
  4. Kim, J.K.; Mastronardi, F.G.; Wood, D.D., et al. Multiple Sclerosis: An Important Role for Post-Translational Modifications of Myelin Basic Protein in Pathogenesis*. Molecular & Cellular Proteomics 2003, 2, 453-462, doi:https://doi.org/10.1074/mcp.M200050-MCP200.
  5. Yang, L.; Tan, D.; Piao, H. Myelin Basic Protein Citrullination in Multiple Sclerosis: A Potential Therapeutic Target for the Pathology. Neurochemical research 2016, 41, 1845-1856, doi:10.1007/s11064-016-1920-2.
  6. Caprariello, A.V.; Rogers, J.A.; Morgan, M.L., et al. Biochemically altered myelin triggers autoimmune demyelination. Proc Natl Acad Sci U S A 2018, 115, 5528-5533, doi:10.1073/pnas.1721115115.
  7. Raijmakers, R.; Vogelzangs, J.; Croxford, J.L., et al. Citrullination of central nervous system proteins during the development of experimental autoimmune encephalomyelitis. The Journal of comparative neurology 2005, 486, 243-253, doi:10.1002/cne.20529.
  8. Darrah, E.; Andrade, F. Rheumatoid arthritis and citrullination. Curr Opin Rheumatol 2018, 30, 72-78, doi:10.1097/bor.0000000000000452.
  9. Tseng, C.C.; Chang, S.J.; Tsai, W.C., et al. Increased incidence of rheumatoid arthritis in multiple sclerosis: A nationwide cohort study. Medicine (Baltimore) 2016, 95, e3999, doi:10.1097/md.0000000000003999.
  10. Ramesh, M.; Gopinath, P.; Govindaraju, T. Role of Post-translational Modifications in Alzheimer's Disease. ChemBioChem 2020, 21, 1052-1079, doi:https://doi.org/10.1002/cbic.201900573.
  11. Shin, M.-K.; Vázquez-Rosa, E.; Koh, Y., et al. Reducing acetylated tau is neuroprotective in brain injury. Cell 2021, 184, 2715-2732.e2723, doi:https://doi.org/10.1016/j.cell.2021.03.032.

Round 2

Reviewer 1 Report

no